# CrossMatch: Cross-Classifier Consistency Regularization for Open-Set Single Domain Generalization

**Ronghang Zhu, Sheng Li**
University of Georgia
{ronghangzhu, sheng.li}@uga.edu

## Abstract

Single domain generalization (SDG) is a challenging scenario of domain generalization, where only one source domain is available to train the model. Typical SDG methods are based on the adversarial data augmentation strategy, which complements the diversity of source domain to learn a robust model. Existing SDG methods require the source and target domains to have the same label space. However, as target domains may contain novel categories unseen in source label space, this assumption is not practical in many real-world applications. In this paper, we propose a challenging and untouched problem: *Open-Set Single Domain Generalization* (OS-SDG), where target domains include unseen categories out of source label space. The goal of OS-SDG is to learn a model, with only one source domain, to classify a target sample with correct class if it belongs to source label space, or assign it to unknown classes. We design a *CrossMatch* approach to improve the performance of SDG methods on identifying unknown classes by leveraging a multi-binary classifier. CrossMatch generates auxiliary samples out of source label space by using an adversarial data augmentation strategy. We also adopt a consistency regularization on generated auxiliary samples between multi-binary classifiers and the model trained by SDG methods, to improve the model's capability on unknown class identification. Experimental results on benchmark datasets prove the effectiveness of CrossMatch on enhancing the performance of SDG methods in the OS-SDG setting.

## 1 Introduction

Deep neural networks have obtained remarkable success in many classification tasks (Sze et al., 2017), as they can learn discriminative feature representations from data. These achievements are strongly relied on the independent and identically distributed (i.i.d) assumption (Vapnik, 1992), i.e., the training and testing samples are from the same distribution. However, in real-world application, the i.i.d assumption is often violated due to the dataset shift problem. As a result, the performance of deep neural networks degrades notably (Hendrycks & Dietterich, 2019; Recht et al., 2019). To tackle this challenge, domain adaptation (DA) (Long et al., 2018; Peng et al., 2020; Zhu et al., 2021b;a) is proposed, which aims to mitigate the discrepancy between the source and target domains. DA assumes that both the source and target domains are accessible during model training stage. Even though DA has demonstrated promising performance on dealing with the dataset shift problem, it still fails in some practical scenarios when the target domains are unaccessible at the model training stage.

Domain generalization (DG) (Zhao et al., 2020b; Zhou et al., 2021b) is introduced to deal with the dataset shift problem and the absence of target domains. The goal of DG is to train a model from multiple source domains comprising the same label space, such that the learned model can generalize well to any unseen target domains. Despite of the success of DG in many real-world applications (Zhou et al., 2021a), it yet fails to a worst-case domain generalization scenario, i.e., single domain generalization (SDG). In SDG, only one source domain is available to train the model while the learned model need to generalize well on many unseen target domains. Several recent studies have made significant progress on this challenging yet practical scenario (Volpi et al., 2018; Zhao et al., 2020a; Qiao et al., 2020; Fan et al., 2021).

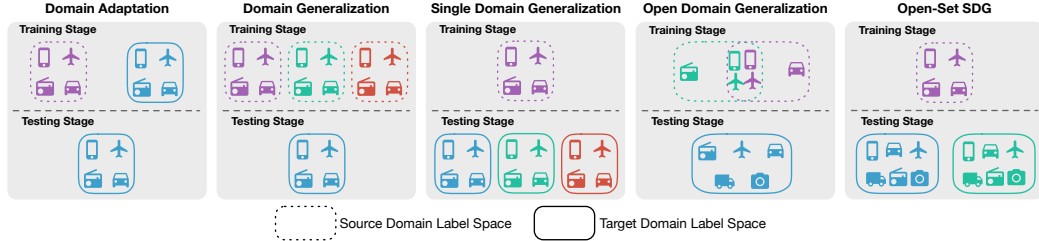

Figure 1: Illustration of label space of domain adaptation (DA), domain generalization (DG), single domain generalization (SDG), open domain generalization (ODG), and open-set single domain generalization (OS-SDG). Different colors represent different domains. As there is no prior knowledge about the target domains, existing DG and SDG methods cannot handle unknown classes (e.g., camera and truck) in target domain. ODG provides multiple source domains to learn a model. However, our proposed OS-SDG only supplies a single source domain, which is more challenging than ODG.

Table 1: Illustration of difference between the proposed Open-Set Single Domain Generalization (OS-SDG) and other related tasks. $\mathcal{D}_s$ and $\mathcal{D}_t$ denote source and target domains, respectively. $\mathcal{C}_s$ and $\mathcal{C}_t$ denote the label space of source and target domains, respectively.

| Task | More than one $\mathcal{D}_s$ for training ? | Need $\mathcal{C}_s = \mathcal{C}_t$? | Access to $\mathcal{D}_t$? |
|---|---|---|---|
| Domain Adaptation | ✗ | ✓ | ✓ |
| Domain Generalization | ✓ | ✓ | ✗ |
| Single Domain Generalization | ✗ | ✓ | ✗ |
| Open-Set Domain Adaptation | ✗ | ✗ | ✓ |
| Open Domain Generalization | ✓ | ✗ | ✗ |
| The Proposed OS-SDG | ✗ | ✗ | ✗ |

One assumption in DG and SDG is that the label spaces of source and target domains are uniform. However, in some more realistic scenarios such as autonomous driving and biomedical applications, it is possible that the target domains contain novel categories that are unseen in the source label space. As a result, the learned model cannot tackle target samples from novel categories, which dramatically degrades the robustness of existing DG and SDG methods. Until recently, this interesting yet tough problem is investigated by Shu et al. (2021), which proposes a task called open domain generalization (ODG). ODG assumes that both source and target domains have different label spaces, and multiple source domains are accessible for training. An ODG model either classifies a target sample with the correct class if it belongs to the multiple source label spaces, or marks it as unknown class.

In this paper, we consider a worst-case open-set problem (Geng et al., 2020) in SDG: only one source domain is accessible for training, and multiple target domains include samples that do not belong to any class in the source label space. We name this novel and challenging problem as *Open-Set Single Domain Generalization* (OS-SDG). Figure 1 illustrates the relationship between source and target label spaces in DA, DG, SDG, ODG, and OS-SDG. We also summarize how our problem is different from several related problems in Table 1. The goal of OS-SDG is to learn a model from a single source domain, which can generalize to unseen target domains with samples from unknown classes.

Inspired by universal domain adaptation (You et al., 2019; Saito et al., 2020; Zhu & Li, 2021), an "unknown" target sample tends to have a larger entropy of the learned model's output than samples from "known" classes. We can draw a boundary between "known" and "unknown" classes by using the entropy of the learned model's outputs. We evaluate the performance of representative SDG methods (Volpi et al., 2018; Zhao et al., 2020a) in the proposed OS-SDG task. Results (refer to Figure 5 in Section 4) show that the existing SDG methods have very limited capability on identifying unknown classes in target domains, although they obtain promising results in terms of per-class mean accuracy. These SDG methods only focus on synthesizing samples from diverse distributions to enrich source domain, and thus they cannot properly identify samples from unknown classes. The research challenge of modeling and identifying unknown classes in open-set single domain generalization is still untouched.

We propose a novel approach named *CrossMatch* to address the OS-SDG problem. First, CrossMatch induces unknown classes information by generating auxiliary samples that are potentially out of the source label space. These auxiliary samples may not belong to the actual unknown classes in the target domains, as we do not have access to the target label space during training stage. However,

we encourage the auxiliary samples to be far away from the known classes in the source domain, and thus they could still assist in identifying whether a sample belongs to known classes or not. In particular, we adopt a multi-binary classifier (Liu et al., 2019; Saito & Saenko, 2021) that consists of multiple one-vs-all binary classifiers. For a given sample, if all the binary classifiers mark it as negative, we believe this sample has a high probability of belonging to an unknown class. Therefore, this characteristic allows us to design a new loss function to generate auxiliary samples for unknown classes. Second, with the generated auxiliary samples for unknown classes, we minimize their entropy of multi-binary classifier's output to help model learn discriminative feature distributions between known and unknown classes, and improve the capability of multi-binary classifier on unknown class identification. Last, we design a novel consistency regularization to further improve the capability of model on unknown class identification by propagating unknown class information, learned by multi-binary classifier, to the model.

## 2 RELATED WORK AND BACKGROUND

**Domain Generalization (DG)** aims to learn a model from multiple source domains and expects it to generalize over unseen target domains. Muandet et al. (2013) proposed a kernel-based method to learn domain invariant feature representations by minimizing the dissimilarity across source domains. Shankar et al. (2018) augment source domain by using adversarial gradients obtained from a domain discriminator. Zhou et al. (2020) synthesize samples by mapping source samples to pseudo-novel domains with an optimal transport-based distance measure. Recently, meta-learning has involved in DG problem (Li et al., 2018; Dou et al., 2019; Du et al., 2020). The main idea lies in learning the model under a domain shift which is induced by meta-train and meta-test from source domains.

**Single Domain Generalization (SDG)** is a more challenging and realistic problem than DG. It assumes the access to a labeled source domain for training, and many unseen target domains for test. The source and target domains are sampled from different distributions. The goal of SDG is to learn a model from the source domain, which can generalize well on many unseen target domains. Adversarial gradient-based augmentation is usually used in SDG for generating new samples to enrich the diversity of source domain. For instance, Adversarial Data Augmentation (ADA) (Volpi et al., 2018) adopts sign-flipped gradients back-propagated from label classifier to generate new samples. Maximum-Entropy Adversarial Data Augmentation (MEADA) (Zhao et al., 2020a) further improves ADA by generating more "hard" samples. Qiao et al. (2020) use an auxiliary Wasserstein autoencoder to help ADA generate more "challenging" samples. Fan et al. (2021) design an adaptive normalization scheme with ADA to enhance the generalization of learned model.

**Open-Set Recognition (OSR)** aims to identify unknown classes samples that are completely unseen during the training stage (Geng et al., 2020). Bendale & Boult (2016) propose a model layer, i.e., OpenMax, to estimate the probability of a test sample belonging to the unknown class. Padhy et al. (2020) apply the one-vs-all classifier to identify unknown class samples. Although many OSR methods have obtained good performance under a strong assumption that the source and target domains have similar distributions, they still fail to address the proposed SS-SDG problem, where the source and target domains are selected from different distributions.

**Open-Set Domain Adaptation (OSDA)** is proposed by Panareda Busto & Gall (2017) where both source and target domains have private label spaces, respectively, and the common label space is known. Recently, Saito et al. (2018) adjust the OSDA setting by claiming no source private label space, which means target label space contains source label space. Recent OSDA (Liu et al., 2019; Bucci et al., 2020) methods focus on this challenging setting. However, OSDA faces two limitations. First, it relies on an assumption that the target domain is accessible during training stage. Second, OSDA methods focus on the scenario that there are only one source domain and one target domain.

**Open Domain Generalization (ODA)** is proposed by (Shu et al., 2021). In this problem, a target domain and multiple source domains have different label spaces. The goal is to learn a model from multiple source domains to correctly classify each target sample into either a known class in the source label space or an unknown class. They utilize meta-learning framework to learn generalizable representations across domains augmented on both feature-level and label-level. Our OS-SDG problem is more challenging than the ODA problem, as OS-SDG has only one source domain for training but multiple target domains for test, while ODA has multiple source domains for training and only one target domain for test.

**Worst-Case Problem** is proposed by (Sinha et al., 2018). In single domain generalization, we consider to learn the model with source domain by solving the following objective function:

$$\min_\theta \sup_{\mathcal{D}_t} \left\{ \mathbb{E}[\mathcal{L}_{ce}(\theta; \mathcal{D}_t) : D(\mathcal{D}_t, \mathcal{D}_s) \leq \rho] \right\}, \tag{1}$$

where $D$ represents a distance metric that measures the similarity between source and target domains. $\rho$ indicates the largest transportation cost of moving source domain $\mathcal{D}_s$ to target domain $\mathcal{D}_t$. $\theta$ denotes model parameters optimized by cross-entropy loss function $\mathcal{L}_{ce}$.

**Adversarial Data Augmentation** (Volpi et al., 2018) aims to enrich the diversity of source domain by generating a domain $\mathcal{D}$ to approximate the unseen target domains $\mathcal{D}_t$. It reformulates worst-case problem (in Eq. 1) into a Lagrangian optimization problem with a fixed penalty parameter $\gamma \geq 0$:

$$\min_\theta \sup_{\mathcal{D}} \left\{ \mathbb{E}[\mathcal{L}_{ce}(\theta; \mathcal{D})] - \gamma D(\mathcal{D}, \mathcal{D}_s) \right\}. \tag{2}$$

Here $D$ denotes the Wasserstein metric (Volpi et al., 2018) used to preserve the semantics of the generated samples. The overall loss function is formulated as:

$$\mathcal{L}_{ada} = \mathcal{L}_{ce}(\theta; \mathcal{D}_s) - \gamma \mathcal{L}_{const}(\theta_g; \mathcal{D}, \mathcal{D}_s), \tag{3}$$

where $\mathcal{L}_{const}(\theta_g; \mathcal{D}, \mathcal{D}_s) = \|G(x) - G(x_s)\|_2^2 + \infty \cdot \mathbf{1}\{y \neq y_s\}$ and $G$ is the feature extractor. The whole training process is an iterative procedure where two stages are alternated, i.e., maximization stage and minimization stage. At maximization stage, the new domain $\mathcal{D}$ is generated from $\mathcal{D}_s$ by maximizing $\mathcal{L}_{ada}$ with learning rate $\eta$:

$$x_{t+1} \leftarrow x_t + \eta \nabla_{x_t} \mathcal{L}_{ada}(\theta; x_t). \tag{4}$$

After the maximization stage, the generated domain $\mathcal{D}$ is appended to $\mathcal{D}_s$. At the minimization stage, $\theta$ is optimized by minimizing the loss $\mathcal{L}_{ce}$ with $\mathcal{D}_s$.

**Multi-Binary Classifier** has proved its capability on unknown class identification in domain adaptation related problems (Liu et al., 2019; Saito & Saenko, 2021). A multi-binary classier $F_b$ with $k$ classes is defined as $F_b = \{F_b^1, \ldots, F_b^k\}$, where $F_b^i$ is the $i$-th one-vs-all binary classifier with output $p_b^i = F_b^i(G(x)) \in \mathbb{R}^2$. We use $p_b^i(t = 0|x)$ and $p_b^i(t = 1|x)$, where $p_b^i(t = 0|x) + p_b^i(t = 1|x) = 1$, to indicate how likely the sample belongs to the $i$-th class and other classes, respectively. Recently, Saito & Saenko (2021) proposes a hard negative binary classifier sampling strategy to help $F_b$ learn a better boundary to identify unknown classes by minimizing the following loss function:

$$\mathcal{L}_{ova}(x, y) = -\log(p_b^y(t = 0|x)) - \min_{i \neq y} \log(1 - p_b^i(t = 1|x)). \tag{5}$$

Here the objective is to improve the robustness of multi-binary classifier on unknown class identification by optimizing each binary-classifier with the corresponding hard negative samples. Furthermore, for unlabeled samples, minimizing their multi-binary entropy loss $\mathcal{L}_{ent}^b$:

$$\mathcal{L}_{ent}^b(x) = -\sum_{i=1}^{k} p_b^i(t = 0|x) \log p_b^i(t = 0|x) + p_b^i(t = 1|x) \log p_b^i(t = 1|x). \tag{6}$$

They will either be aligned to source samples or be kept as unknown, and help improve the confidence of model on unknown class identification.

## 3 METHODOLOGY

In this section, we formally introduce the Open-Set Single Domain Generalization (OS-SDG) task, and then describe the proposed CrossMatch approach in detail.

### 3.1 OPEN-SET SINGLE DOMAIN GENERALIZATION

Given a labeled source domain $\mathcal{D}_s = \{(x_s^i, y_s^i)\}_{i=1}^{n_s}$ with label space $\mathcal{C}_s$, and multiple unseen target domains $\mathcal{D}_t = \{\mathcal{D}_t^1, \ldots, \mathcal{D}_t^n\}$ where $\mathcal{D}_t^i = \{(x_{t,i}^j, y_{t,i}^j)\}_{j=1}^{n_t^i}$ with label space $\mathcal{C}_t = \mathcal{C}_t^1 = \cdots = \mathcal{C}_t^n$. The source and target domains are sampled from different distributions. As target domains have novel classes that do not belong to $\mathcal{C}_s$, we denote those novel classes as an unknown class space in target

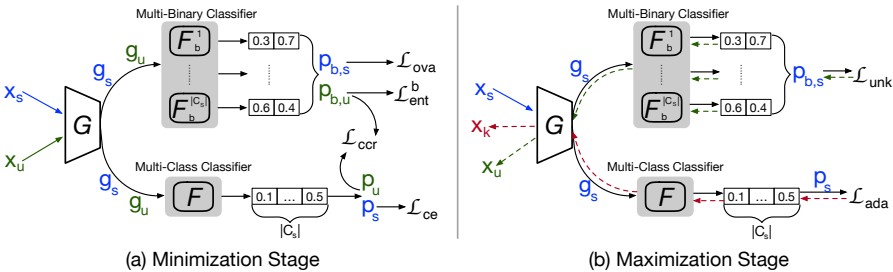

Figure 2: Overview of CrossMatch in SDG methods. $x_s$ represents source samples while $x_k$ and $x_u$ denote generated known class and auxiliary unknown class samples, respectively. $G$ is the feature extractor where $g_s = G(x_s)$. The multi-binary classifier is $F_b = \{F_b^1, \ldots, F_b^{|\mathcal{C}_s|}\}$ where $p_s^b = F_b(g_s)$. $F$ is the multi-class classifier where $p_s = F(g_s)$. (a) At minimization stage, we adopt $\mathcal{L}_{ova}$(Eq. 5) and $\mathcal{L}_{ent}^b$ (Eq. 6) to optimize $F_b$ while train $F$ by cross-entropy loss $\mathcal{L}_{ce}$. We also propose a novel loss $\mathcal{L}_{ccr}$ to improve the capability of $F$ on unknown class identification. (b) At maximization stage, SDG methods maximize their defined losses, e.g., $\mathcal{L}_{ada}$ (Eq. 3), to generate new samples to enrich the diversity of source domain in $\mathcal{C}_s$, while we propose an auxiliary sample generation loss $\mathcal{L}_{unk}$ to generate samples that are far way from $\mathcal{C}_s$. After maximization stage, the generated $x_k$ is appended to source domain $\mathcal{D}_s$.

domains, $\mathcal{C}_t^u = \mathcal{C}_t \backslash \mathcal{C}_s$. The goal of OS-SDG is to train a model with $\mathcal{D}_s$ to classify target samples into $|\mathcal{C}_s| + 1$ classes, where the novel classes in target domains are treated as one unknown class. Figure 1 illustrates the relationship of label space between source and target domains in OS-SDG. The research challenges in OS-SDG are twofold. First, how can we generate unknown classes samples that could complement sample diversity in the source domain? Second, what is an effective strategy to improve the model's capability on unknown class identification?

## 3.2 CROSSMATCH

The proposed CrossMatch approach generates auxiliary samples for unknown classes out of $\mathcal{C}_s$ (Sec. 3.2.1) and then exploits those samples to improve the capability of existing SDG methods on unknown class identification (Sec. 3.2.2). An overview of the proposed CrossMatch approach is shown in Figure 2.

Inspired by the worst-case problem defined in SDG, we extend it to OS-SDG setting based on the CrossMatch approach, as follows:

$$\min_{\theta, \theta_{f_b}} \sup_{\mathcal{D} = \{\mathcal{D}_k, \mathcal{D}_u\}} \left\{ \mathbb{E}[\mathcal{L}_k(\theta, \theta_{f_b}; \mathcal{D}_s) + \mathcal{L}_u(\theta, \theta_{f_b}; \mathcal{D}_u) + \mathcal{L}_{unk}(\mathcal{D}_s)] : D(\mathcal{D}_k, \mathcal{D}_s) \leq \rho, D(\mathcal{D}_u, \mathcal{D}_s) \geq \rho \right\},$$
(7)

where $\theta = \{\theta_g, \theta_f\}$ includes the parameters of feature extractor $G$ and multi-class classifier $F$. $\theta_{f_b}$ is the parameters of multi-binary classifier $F_b$. $\mathcal{D} = \{\mathcal{D}_k, \mathcal{D}_u\}$ denotes the augmented domain consists of known classes domain $\mathcal{D}_k$ and auxiliary unknown classes domain $\mathcal{D}_u$. $\mathcal{L}_k = \{\mathcal{L}_{ce} + \mathcal{L}_{ova}\}$ represents those loss functions who use $\mathcal{D}_s$ to optimize model, i.e., cross-entropy loss $\mathcal{L}_{ce}(\theta_g, \theta_f; x_s)$ and one-vs-all loss $\mathcal{L}_{ova}(\theta_g, \theta_{f_b}; x_s)$ (in Eq. 5). $\mathcal{L}_u = \{\mathcal{L}_{ent}^b + \alpha\mathcal{L}_{ccr}\}$ includes two loss functions that optimize model with $\mathcal{D}_u$, i.e., multi-binary entropy loss $\mathcal{L}_{ent}^b(\theta_g, \theta_{f_b}; x_u)$ (Eq. 6) and the proposed cross-classifier consistency regularization loss $\mathcal{L}_{ccr}(\theta_g, \theta_f, \theta_{f_b}; x_u)$ that improve the capability of $F$ on unknown class identification. $\mathcal{L}_{unk}(\mathcal{D}_s)$ (Eq. 13) is a newly proposed loss function for generating auxiliary samples. $D$ is the Wasserstein metric and $\rho$ is the largest domain discrepancy between two domains.

**At minimization stage**, the overall loss function is defined as follows:

$$\mathcal{L}_{min} = \mathcal{L}_{ce} + \mathcal{L}_{ova} + \mathcal{L}_{ent}^b + \alpha\mathcal{L}_{ccr},$$
(8)

where $\alpha$ is the hyper-parameter.

**At maximization stage**, we aim to generate two kinds of samples: known class samples $x_k \in \mathcal{D}_k$ and auxiliary samples $x_u \in \mathcal{D}_u$ which are far away from the known classes in the source label space. For known class samples, we consider the following Lagrangian relaxation derived from Eq. 7 with

penalty parameter $\gamma$:

$$\sup_{\mathcal{D}_k} \left\{ \mathbb{E}[\mathcal{L}_{ce}(\theta; \mathcal{D}_k)] - \gamma D(\mathcal{D}_k, \mathcal{D}_s) \right\}, \tag{9}$$

where $D(\mathcal{D}_k, \mathcal{D}_s) = \mathcal{L}_{const}(\theta_g; \mathcal{D}_k, \mathcal{D}_s)$ is defined in Eq. 3. To solve this penalty problem in Eq. 9, we adopt the same strategy in (Volpi et al., 2018) with the suitable conditions (Boyd et al., 2004) by performing stochastic gradient descent procedures. We define the robust surrogate loss $\phi(\theta; x_s, y_s) = \sup_{x_k \in \mathcal{D}_k} \{\mathcal{L}_{ce}(\theta; x_k, y_s) - \gamma \mathcal{L}_{const}(\theta_g; x_k, x_s)\}$ and have $\nabla_\theta \phi(\theta; x_s, y_s) = \nabla_\theta \mathcal{L}_{ce}(\theta; x_k^*, y_s)$, where $x_k^* = \arg\max_{x_k \in \mathcal{D}_k} \{\mathcal{L}_{ce}(\theta; x_k, y_s) - \gamma \mathcal{L}_{const}(\theta_g; x_k, x_s)\}$ is an adversarial perturbation of $x_s$ at the current model $\theta$. We adopt a few iterations to generate known class samples with learning rate $\eta$ by:

$$x_k^{t+1} \leftarrow x_k^t + \eta \nabla_{x_k^t} \{\mathcal{L}_{ce}(\theta; x_k^t, y_s) - \gamma \mathcal{L}_{const}(\theta_g; x_k^t, x_s)\}. \tag{10}$$

The generation procedure in Eq. 10 is consistent with the single domain generation method ADA (Volpi et al., 2018), which can be further extended to other single domain generation methods, e.g., MEADA (Zhao et al., 2020a), by adding more loss objectives.

Similar to the way of generating known class samples, we also convert Eq. 7 into a Lagrangian relaxation to generate auxiliary samples with penalty parameter $\gamma$:

$$\sup_{\mathcal{D}_u} \left\{ \mathbb{E}[\mathcal{L}_{unk}(\mathcal{D}_u)] + \gamma D(\mathcal{D}_u, \mathcal{D}_s) \right\}. \tag{11}$$

By adopting the same strategy as known class sample generation process, we can generate auxiliary samples with the following rule:

$$x_u^{t+1} \leftarrow x_u^t + \eta \nabla_{x_u^t} \{\mathcal{L}_{unk}(x_u^t) + \gamma \mathcal{L}_{const}(\theta_g; x_u^t, x_s)\}, \tag{12}$$

where $\eta$ is the learning rate of stochastic gradient ascent. The details of $\mathcal{L}_{unk}$ and $\mathcal{L}_{ccr}$ will be described in Section 3.2.1 and Section 3.2.2, respectively.

**Remarks.** CrossMatch aims to tackle the challenges in OS-SDG from two aspects: (i) generation of auxiliary samples for unknown classes, and (ii) improvement of unknown class identification. The key innovation of the proposed approach lies in jointly encouraging the model to generate auxiliary samples for unknown classes by maximizing the newly designed loss $\mathcal{L}_{unk}$ at the maximization stage, and improving the capability of multi-class classifier $F$ on unknown class identification. In particular, we optimize the proposed cross-classifier consistency regularization loss $\mathcal{L}_{ccr}$ with generated auxiliary samples for unknown classes at the minimization stage.

### 3.2.1 Auxiliary Sample Generation for Unknown Classes

The key idea of adversarial data augmentation is to use sign-flipped gradients back-propagated from a label classifier or other related terms (Volpi et al., 2018; Qiao et al., 2020) to generate diversified samples. Inspired by the structure of multi-binary classifier $F_b$, if a given sample is classified as "other class" by all the binary classifiers in $F_b$, it has a high probability to be associated with unknown classes. For example, $\mathcal{C}_s$ contains four known classes (e.g., "phone", "airplane", "radio" and "car") and $\mathcal{C}_t^u$ consists of two unknown classes "truck" and "camera", as illustrated in Figure 1. We assume that $F_b$ has been well trained in the source domain. If a target sample belongs to the class "phone", the corresponding binary classifier should output a high probability to "phone", while the other binary classifiers will classify it as "other class". But, if a target sample belongs to "truck", all the binary classifiers would classify it as "other class".

With this insight, we design a novel auxiliary sample generalization loss $\mathcal{L}_{unk}$ to generate samples that are possibly out of source label space. Given a source sample $x_s$ with label $y_s$, multi-binary classifier $F_b$ and feature extractor $G$, we denote $\mathcal{L}_{unk}$ as:

$$\mathcal{L}_{unk}(x_s, y_s) = -\log(p_{b,s}^{y_s}(t=0|x_s)) + \sum_{i \neq y_s}^{k} \log p_{b,s}^{i}(t=1|x_s), \tag{13}$$

where $p_{b,s}^{y_s}(t=0|x_s)$ and $p_{b,s}^{y_s}(t=1|x_s)$ represent the probabilities of being $y_s$-th class and other classes from the $y_s$-th binary classifier, respectively. By maximizing $\mathcal{L}_{unk}$ at the maximization stage, it encourages the model to generate new samples that all binary classifiers in $F_b$ predict it as "other class". Thus, the new generated sample could belong to unknown classes. As entropy loss function has proved its effectiveness on diversifying the adversarial generated samples (Zhao et al., 2020a), to further enrich the diversity of auxiliary samples, we also incorporate the multi-binary loss $\mathcal{L}_{ent}^{b}$ and one-vs-all loss $\mathcal{L}_{ova}$ to $\mathcal{L}_{unk}$.

### 3.2.2 CROSS-CLASSIFIER CONSISTENCY REGULARIZATION

With the generated auxiliary samples for unknown classes, we minimize the multi-binary entropy loss $\mathcal{L}_{ent}^b$ to enhance the capability of multi-binary classifier $F_b$ on unknown class identification, and help feature extractor $G$ learn discriminative feature representations, which benefit the multi-class classifier $F$ on unknown class identification. Each auxiliary sample $x_u$ will be aligned to source samples if it belongs to $\mathcal{C}_s$, or kept as unknown. To make full use of unknown classes information learned by $F_b$, we propose the cross-classifier consistency regularization (CCR) loss to propagate those information learned by $F_b$ to $F$. The key idea lies in encouraging $F_b$ and $F$ to produce the similar output distribution, in form of one-vs-all, for each generated auxiliary sample from $\mathcal{D}_u$:

$$\mathcal{L}_{ccr}(x_u) = \sum_{i=1}^{k} \left\| p_{b,u}^i - p_{b',u}^i \right\|_2^2, \tag{14}$$

where $p_{b,u}^i \in \mathbb{R}^2$ represents the $i$-th binary classifier's output. $p_{b',u}^i$ indicates the one-vs-all version of multi-class classifier $F$'s output $p_u$ for $i$-th class, i.e., $p_{b',u}^i = \left[ p_u^i, 1 - p_u^i \right]$, $p_u^i$ is the probability of $x$ belongs to $i$-th class. Different from previous consistency regularization (Laine & Aila, 2017; Sajjadi et al., 2016), which encourages the classifier to produce similar outputs for sample under different augmentations, our CCR encourages different classifiers to generate similar output distributions for a given sample.

### 3.3 INFERENCE

Figure 3 illustrates the inference procedure where the learned model $M$ consists of a feature extractor $G$ and a multi-class classifier $F$. The target sample is marked as unknown class if its entropy is larger than the threshold $\mu$. Otherwise, it will be assigned to a class in the source label space $\mathcal{C}_s$.

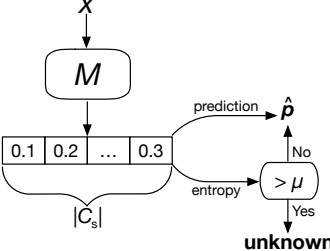

Figure 3: Inference procedure.

## 4 EXPERIMENTS

### 4.1 EXPERIMENTAL SETUP

We briefly introduce the experimental setup in this section. More details about the experimental settings and experimental results are provided in the Appendix due to space limit.

**Datasets.** (1) ***Digits*** comprises of five digits datasets: **MNIST** (LeCun et al., 1989), **SVHN** (Netzer et al., 2011), **USPS** (Hull, 1994), **MNIST-M** and **SYN** (Ganin & Lempitsky, 2015). **MNIST** is the source domain, and its label space includes numbers from 0 to 4. The other datasets are treated as target domains, and their target unknown label space $\mathcal{C}_t^u$ contains numbers from 5 to 9. (2) ***Office31*** (Saenko et al., 2010) contains 31 classes collected from three visually distinct domains: Amazon, DSLR and Webcam. The 10 classes shared by Office-31 and Caltech-256 (Gong et al., 2012) form the source label space $\mathcal{C}_s$. In alphabetical order, the last 11 classes are used as target unknown class space $\mathcal{C}_t^u$. Due to DSLR and Webcam are relatively small, we only use Amazon as the source domain. (3) ***Office-Home*** (Venkateswara et al., 2017) consists of four domains: Artistic, Clip Art, Product, and Real-World. It comprises of 65 categories from four dissimilar domains. In alphabetic order, we select the first 15 classes as the source label space $\mathcal{C}_s$. The remaining 50 classes are viewed as target unknown label space $\mathcal{C}_t^u$. Each domain serves as source domain, and the rest are target domains. (4) **PACS** (Li et al., 2017) consists of four domains: Art Paint, Cartoon, Sketch, and Photo. It has 9,991 images from seven object categories.

**Implementation Details.** For ***Digits***, we adopt the ConvNet (LeCun et al., 1989) with architecture *conv-pool-conv-pool-fc-fc-softmax*, resize all images to $32 \times 32$, and follow the settings in Volpi et al. (2018); Zhao et al. (2020a). For ***Office31***, ***Office-Home*** and ***PACS***, we use an ImageNet-pretrained ResNet18 (He et al., 2016) as the base network.

**Baselines and Metrics.** We adopt the Empirical Risk Minimization (ERM) (Koltchinskii, 2011) as a simple baseline without any consideration of SDG. We compare with two state-of-the-art SDG methods, Adversarial Data Augmentation (ADA) (Volpi et al., 2018) and Maximum-Entropy Adversarial Data Augmentation (MEADA) (Zhao et al., 2020a). We also adopt Open Set Domain

Table 2: Results (%) on *Digits* (ConvNet) and *Office31* (ResNet18).

| Dataset | Metric | OSDAP | OpenMax | ERM | +CM | ADA | +CM | MEADA | +CM |
|---|---|---|---|---|---|---|---|---|---|
| Digits | $acc$ | 41.42 | 42.38 | 49.17 | 49.07 | 50.22 | 49.71 | 52.98 | 51.27 |
| | $acc_u$ | 70.60 | 83.81 | 13.04 | **53.52** | 15.11 | **52.07** | 29.83 | **46.11** |
| | $acc_k$ | 35.59 | 34.40 | 56.40 | 48.67 | 57.24 | 49.24 | 57.61 | 52.30 |
| | $hs$ | 40.46 | 40.67 | 17.97 | **40.15** | 20.14 | **39.93** | 30.37 | **38.70** |
| Office31 | $acc$ | 76.51 | 18.19 | 79.82 | 78.30 | 80.13 | 78.61 | 80.26 | 78.98 |
| | $acc_u$ | 84.28 | 100.0 | 27.04 | **37.60** | 25.24 | **34.51** | 25.09 | **41.08** |
| | $acc_k$ | 75.77 | 10.01 | 85.10 | 82.37 | 85.62 | 83.02 | 85.78 | 82.77 |
| | $hs$ | 77.68 | 16.74 | 40.69 | **51.14** | 38.65 | **48.50** | 38.55 | **54.69** |

Table 3: Results (%) on *Office-Home* (ResNet18).

| Method | Artistic | | Clip Art | | Product | | Real-World | | Average | |
|---|---|---|---|---|---|---|---|---|---|---|
| | $acc$ | $hs$ | $acc$ | $hs$ | $acc$ | $hs$ | $acc$ | $hs$ | $acc$ | $hs$ |
| OSDAP | 45.61 | 52.35 | 52.78 | 58.82 | 41.45 | 47.95 | 53.51 | 58.40 | 48.34 | 54.38 |
| OpenMax | 22.42 | 30.64 | 22.67 | 29.51 | 15.10 | 16.65 | 25.54 | 33.07 | 21.43 | 27.47 |
| ERM | 65.00 | 31.07 | 64.12 | 35.78 | 60.53 | 36.33 | 66.59 | 33.92 | 64.06 | 34.28 |
| ERM+CM | 65.49 | 52.85 | 63.37 | 50.51 | 58.03 | 47.25 | 67.75 | 52.60 | 63.66 | **50.80** |
| ADA | 68.29 | 32.94 | 65.10 | 42.09 | 60.52 | 34.72 | 67.04 | 34.86 | 65.24 | 36.15 |
| ADA+CM | 66.30 | 46.68 | 62.64 | 49.31 | 58.72 | 47.47 | 66.82 | 50.47 | 63.62 | **48.48** |
| MEADA | 68.31 | 33.29 | 65.25 | 42.05 | 60.43 | 35.68 | 67.04 | 34.65 | 65.01 | 36.42 |
| MEADA+CM | 65.85 | 53.22 | 62.90 | 48.87 | 58.36 | 45.34 | 67.10 | 50.77 | 63.55 | **49.55** |

Adaptation by Backpropagation (OSDAP) (Saito et al., 2018) a representative method in open-set domain adaptation and OpenMax (Bendale & Boult, 2016) a typical open-set recognition method as baselines. Furthermore, we extend ERM, ADA, and MEADA using the proposed CrossMatch (CM) approach (denoted by "+CM"). We adopt the overall mean accuracy ($acc$), known class accuracy ($acc_k$) unknown class accuracy ($acc_u$) and *h-score* (*hs*) (Fu et al., 2020) as evaluation metrics.

## 4.2 RESULTS AND DISCUSSIONS

**Digits and Office31.** Table 2 shows the results of our approach and baselines on the *Digits* and *Office31* datasets. Our proposed method is superior to ADA by more than 19% and 9% with respect to *hs* on *Digits* and *Office31*, respectively. Compared with the state-of-the-art method MEADA, which obtains the best *acc* among all the methods, our method still significantly outperforms it with more than 8% and 16% in terms of *hs* on *Digits* and *Office31*, respectively. Meanwhile, our method achieves comparable mean accuracy ($acc$) than baselines. The strong baseline method OSDAP, uses source and target domains to learn the model, gets better results with regard to $acc_u$ in *Digits* and *Office31* datasets. While our method learns the model without target domain but gets similar performance in terms of *hs* in *Digits*, which proves the effectiveness of our method. Compared with representative open-set recognition method, OpenMax, our method obtains similar results in terms of *hs* and better performances with regard to $acc$ and $acc_k$ in *Digits* dataset. OpenMax, without considering the domain shift problem, gets high performance in terms of $acc_u$ by wrongly marking most of testing samples as unknown class. When domain shift is larger in *Office31* dataset, we find that OpenMax gets worse performance in terms of $acc$, $acc_k$, and *hs*.

**Office-Home.** Table 3 shows the *acc* and *hs* on *Office-Home*. Each time, one domain serves as the source domain and the rest three ones are target domains. *Office-Home* is more challenging than the previous two datasets because it contains more target unknown classes than them. Although the performance of our method on *acc* is slightly lower than baselines, we observe remarkable improvement of 16.52%, 12.22%, and 13.13% compared with ERM, ADA, and MEADA in terms of *hs*, respectively. Such improvements over baselines demonstrate the effectiveness of our method. Furthermore, compared with strong baseline OSDAP, our method gets better performance in terms of *acc* and similar results on *hs*, demonstrates the efficiency of our method. OpenMax fails to correctly identify known classes due to significant domain shift existed in *Office-Home* dataset.

**PACS.** We further evaluate the effectiveness of our method on *PACS* dataset. As shown in Table 4, our method outperforms all the baselines in terms of *acc* and *hs*. In particular, our method significantly

Table 4: Results (%) on *PACS* (ResNet18).

| Method | Art Paint | | Cartoon | | Sketch | | Photo | | Average | |
|---|---|---|---|---|---|---|---|---|---|---|
| | *acc* | *hs* | *acc* | *hs* | *acc* | *hs* | *acc* | *hs* | *acc* | *hs* |
| OSDAP | 53.30 | 46.58 | 43.73 | 38.81 | 42.05 | 41.03 | 30.81 | 32.89 | 42.47 | 39.83 |
| OpenMax | 52.59 | 53.60 | 31.71 | 25.23 | 29.85 | 19.87 | 27.60 | 19.47 | 35.44 | 29.54 |
| ERM | 62.24 | 38.90 | 55.34 | 40.96 | 39.19 | 28.89 | 38.32 | 35.74 | 48.77 | 36.12 |
| ERM+CM | 63.52 | 44.9 | 57.6 | 48.31 | 38.53 | 30.43 | 42.52 | 41.6 | **50.54** | **41.31** |
| ADA | 62.48 | 39.02 | 56.43 | 41.55 | 39.03 | 26.93 | 40.28 | 38.13 | 49.56 | 36.41 |
| ADA+CM | 64.26 | 42.4 | 60.41 | 51.81 | 42.48 | 35.18 | 43.97 | 42.76 | **52.78** | **43.04** |
| MEADA | 62.43 | 38.85 | 56.1 | 41.34 | 38.89 | 26.43 | 39.88 | 38.24 | 49.33 | 36.22 |
| MEADA+CM | 62.63 | 41.88 | 60.03 | 51.36 | 41.51 | 35.76 | 43.5 | 41.6 | **51.92** | **42.65** |

improves the capability of baselines, i.e., ERM, ADAN and MEADA, on unknown class identification while slightly enhances the averaged accuracy over all classes. For other two baselines, i.e., OSDAP and OpenMax, we observe the similar phenomenons as *Office-Home* when large domain gaps are existed among source and target domains.

## 4.3 ANALYSIS

**On the Effect of $\mathcal{L}_{unk}$ and $\mathcal{L}_{ccr}$.** We conduct ablation studies to examine the effectiveness of the proposed $\mathcal{L}_{unk}$ (Eq. 13) and $\mathcal{L}_{ccr}$ (Eq. 14) and show the results in Table 5. The top row denotes the performance of baseline, i.e., MEADA. Firstly, the middle row demonstrates that simply maximizing the combination of $\mathcal{L}_{ent}^b$ and $\mathcal{L}_{ova}$ to generate "auxiliary" samples for unknown class would slightly decrease the performance of baseline.

Table 5: Ablation study for auxiliary sample generation on *Digits* dataset.

| $\mathcal{L}_{ent}^b + \mathcal{L}_{ova}$ | $\mathcal{L}_{unk}$ | $\mathcal{L}_{ccr}$ | *acc* | $acc_u$ | *hs* |
|---|---|---|---|---|---|
| | | | 52.98 | 29.83 | 30.37 |
| ✓ | | | 51.86 | 29.01 | 29.36 |
| ✓ | ✓ | | 51.08 | 42.55 | 36.94 |
| ✓ | | ✓ | 52.16 | 30.47 | 30.55 |
| ✓ | ✓ | ✓ | 51.27 | 46.11 | 38.70 |

With the proposed $\mathcal{L}_{unk}$, we observe that the performance surpasses the baseline by more than 12% and 6% in terms of $acc_u$ and *hs*, which proves the effectiveness of proposed $\mathcal{L}_{unk}$. Moreover, we observe that incorporating $\mathcal{L}_{ccr}$ can consistently improve the performance of both models in the middle row, which validate the effectiveness of $\mathcal{L}_{ccr}$.

**Varying Size of known classes.** To verify the robustness of our method under different sizes of known classes, we conduct experiments on *Office-Home* where Real-World is selected as the source domain. The size of known classes varies from 10 to 60. As shown in Figure 4, our method continuously outperforms MEADA with significant improvements in terms of *hs* while obtains similar results as MEADA with respect to *acc*. It indicates that our method can consistently improve the capability of SDG methods on unknown class identification.

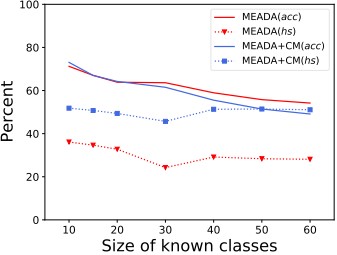

Figure 4: Varying the size of known classes.

## 5 CONCLUSION

In this paper, we propose a new domain generalization task, i.e., open-set single domain generalization (OS-SDG), which aims to generalize the model from a single source domain to unseen target domains and simultaneously deal with unknown classes. We further propose a novel CrossMatch approach to tackle the OS-SDG task, which generates auxiliary samples for unknown classes and improves the capability of unknown class identification with a novel consistency regularization. Extensive experiments demonstrate that the proposed CrossMatch method largely facilitates the capability of existing single domain generalization methods on unknown class identification.

## ACKNOWLEDGEMENT

This research is supported by the U.S. Army Research Office Award (W911NF-21-1-0109).

9

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

## A APPENDIX

The appendix provides descriptions of algorithm, details for datasets and experimental settings, and additional experimental results.

### A.1 ALGORITHM

Algorithm 1 illustrates details of our proposed method.

---

**Algorithm 1** CrossMatch with Adversarial Data Augmentation based Single Domain Generalization methods.

---

**Input:** Source domain $\mathcal{D}_s = \{(x_s^i, y_s^i)\}_{i=1}^{n_s}$ and generated auxiliary unknown class domain $\mathcal{D}_u = \{\}$. Initialized parameters $\theta_g$, $\theta_f$ and $\theta_{f_b}$ for feature extractor $G$, multi-class classifier $F$ and multi-binary classifier $F_b$.

**Output:** Learned feature extractor $G$ and multi-class classifier $F$;

1: **for** $k = 1$ to $K$ **do**          ▷ Run the maximizing procedure $K$ times
2:     **for** $t = 1$ to $T_{MIN}$ **do**          ▷ Run the minimization stage $T_{MIN}$ times
3:        Sample $(x_s, y_s)$ uniformly from $\mathcal{D}_s$
4:        $(\theta_g, \theta_f) \leftarrow (\theta_g, \theta_f) - \beta \nabla_{(\theta_g, \theta_f)} \mathcal{L}_{ce}(\theta_g, \theta_f; x_s, y_s)$
5:        $(\theta_g, \theta_{f_b}) \leftarrow (\theta_g, \theta_{f_b}) - \beta \nabla_{(\theta_g, \theta_{f_b})} \mathcal{L}_{ova}(\theta_g, \theta_{f_b}; x_s, y_s)$
6:        **if** $\mathcal{D}_u$ is not empty **then**
7:           Sample $x_u$ uniformly from $\mathcal{D}_u$
8:           $(\theta_g, \theta_{f_b}) \leftarrow (\theta_g, \theta_{f_b}) - \beta \nabla_{(\theta_g, \theta_{f_b})} \mathcal{L}_{ent}^b(\theta_g, \theta_{f_b}; x_u)$
9:           $(\theta_g, \theta_f, \theta_{f_b}) \leftarrow (\theta_g, \theta_f, \theta_{f_b}) - \beta \nabla_{(\theta_g, \theta_f, \theta_{f_b})} \mathcal{L}_{ccr}(\theta_g, \theta_f, \theta_{f_b}; x_u)$
10:        **end if**
11:     **end for**
12:     **for all** $(x_s, y_s) \in \mathcal{D}_s$ **do**
13:        $x_k \leftarrow x_s$ and $x_u \leftarrow x_s$
14:        **for** $t = 1$ to $T_{MAX}$ **do**          ▷ Run the maximization stage $T_{MAX}$ times
15:           $x_k \leftarrow x_k + \eta \nabla_{x_k} \left\{ \mathcal{L}_{sdg} - \gamma \mathcal{L}_{const}(x_k, x_s) \right\}$
16:           $x_u \leftarrow x_u + \eta \nabla_{x_u} \left\{ \mathcal{L}_{unk} + \mathcal{L}_{ova} + + \mathcal{L}_{ent}^b + \gamma \mathcal{L}_{const}(x_u, x_s) \right\}$
17:           Append $(x_k, y_s)$ to $\mathcal{D}_s$ and $x_u$ to $\mathcal{D}_u$
18:        **end for**
19:     **end for**
20: **end for**
21: **while** not reach maximum steps $T$ **do**
22:     Sample $(x_s, y_s)$ uniformly from $\mathcal{D}_s$
23:     Sample $x_u$ uniformly from $\mathcal{D}_u$
24:     $(\theta_g, \theta_f) \leftarrow (\theta_g, \theta_f) - \beta \nabla_{(\theta_g, \theta_f)} \mathcal{L}_{ce}(\theta_g, \theta_f; x_s, y_s)$
25:     $(\theta_g, \theta_{f_b}) \leftarrow (\theta_g, \theta_{f_b}) - \beta \nabla_{(\theta_g, \theta_{f_b})} \mathcal{L}_{ova}(\theta_g, \theta_{f_b}; x_s, y_s)$
26:     $(\theta_g, \theta_{f_b}) \leftarrow (\theta_g, \theta_{f_b}) - \beta \nabla_{(\theta_g, \theta_{f_b})} \mathcal{L}_{ent}^b(\theta_g, \theta_{f_b}; x_u)$
27:     $(\theta_g, \theta_f, \theta_{f_b}) \leftarrow (\theta_g, \theta_f, \theta_{f_b}) - \beta \nabla_{(\theta_g, \theta_f, \theta_{f_b})} \mathcal{L}_{ccr}(\theta_g, \theta_f, \theta_{f_b}; x_u)$
28: **end while**

---

### A.2 DATASETS

***Digits*** consists of 5 different datasets: MNIST (LeCun et al., 1989), SVHN (Netzer et al., 2011), USPS (Hull, 1994), SYN and MNIST-M (Ganin & Lempitsky, 2015). Following the setting defined by (Volpi et al., 2018; Zhao et al., 2020a), we select 10,000 images from MNIST as the source domain and the rest datasets are viewed as target domains. The source label space $\mathcal{C}_s$ contains numbers from 0 to 4 and target unknow class space $\mathcal{C}_t^u$ includes numbers from 5 to 9. We resize all the images to 32×32 and duplicate their channels to convert all the grayscale images to RGB.

***Office31*** (Saenko et al., 2010) includes 4,652 images in 31 categories collected from three domains: Amazon, Webcam and DSLR. We follow the protocol proposed by (Panareda Busto & Gall, 2017). We use the 10 classes (back pack, bike, calculator, headphones, keyboard, laptop, monitor, mouse,

mug and projector) shared by Office-31 and Caltech-256 (Gong et al., 2012) as the source domain label space $\mathcal{C}_s$. Then in alphabetical order, the last 11 classes (ruler, punchers, stapler, scissors, trash can, tape dispenser, pen, phone, printer, ring binder and speaker) are used as target unknown class space $\mathcal{C}_t^u$. Due to the number of images in DSLR and Webcam are relatively small, we only use Amazon as the source domain.

***Office-Home*** (Venkateswara et al., 2017) contains 15,500 images from four domains: Artistic, Clip Art, Product, and Real-World. As it comprises of 65 categories from four dissimilar domains, it's more challenging than *Digits* and *Office-31*. In alphabetic order, we select the first 15 classes (alarm clock, backpack, battery, bed, bike, bottle, bucket, calculator, calendar, candles, chair, clipboards, computer, couch and curtains) as the source label space $\mathcal{C}_s$. The remaining 50 classes are viewed as target unknown label space $\mathcal{C}_t^u$. Each time, one domain is selected as source domain and the rest are viewed as target domain.

***PACS*** (Li et al., 2017) is a recent challenging benchmark for domain generalization that includes four domains: Art Paint, Cartoon, Sketch, and Photo. It consists of 9,991 images and 7 object categories shared across these domains. In alphabetic order, we choose the first four classes (dog, elephant, giraffe, and guitar.) as the source label space $\mathcal{C}_s$. The last three classes (horse, house, and person.) are selected as target unknown label space $\mathcal{C}_t^u$. We train the model by one domain and evaluate the learned model by the rest domains

### A.3 EVALUATION METRICS.

We adopt the evaluation protocol in Open-Set Domain Adaptation (Panareda Busto & Gall, 2017), where all the target classes out of $\mathcal{C}_s$ are regarded as one unified unknown class. We compute the mean accuracy averaged over all classes in $|\mathcal{C}_s| + 1$ (i.e., $acc$) and the unknown class accuracy (i.e., $acc_u$) across all target domains. To better evaluate the capability of methods on unknown class identification, we also adopt the *h-score* (*hs*) (Fu et al., 2020) metric, $hs = 2 * acc_s * acc_u / (acc_s + acc_u)$. It is the harmonic mean of average per-class accuracy in $\mathcal{C}_s$ (i.e., $acc_k$) and $acc_u$. The *h-score* value is high only when both $acc_k$ and $acc_u$ are high.

### A.4 IMPLEMENTATION DETAILS

We set the same parameters and optimization functions to baselines and ours.

**Digits.** Following the setup of (Volpi et al., 2018; Zhao et al., 2020a). We use ConvNet (LeCun et al., 1989) (*conv-pool-conv-pool-fc-fc-softmax*) as the base model. The batch size is 32. We adopt Adam with learning rate $\beta = 0.0001$ for minimization stage and SGD with learning rate $\eta = 1.0$ for maximization stage. We set $T = 10000$, $T_{MIN} = 100$, $T_{MAX} = 15$, $K = 3$, $\gamma = 1.0$ and $\alpha = 1$. The reported results are the averaged results based on 10 random experiments.

**Office31, Office-Home and PACS.** Referring to the setup of (Zhao et al., 2020a) on *PACS*, we adopt ImageNet-pretrained ResNet18 as base network and set the batch size to 32. We use SGD with learning rate $\beta = 0.001$ which adjusted by a cosine annealing schedule (Zagoruyko & Komodakis, 2016), the momentum of 0.9 and the weight decay of 0.00005 for minimization stage. We adopt the SGD with learning $\eta = 1.0$ for maximization stage. We set $T = 10000$, $T_{MIN} = 100$, $T_{MAX} = 15$, $K = 1$, $\gamma = 1.0$ and $\alpha = 1$. The reported results are the averaged results based on 3 random experiments.

### A.5 CONVERGENCE ANALYSIS

We testify the convergence performance of our method under $acc$ and $acc_u$ by comparing it against ADA and MEAD on the *Digits* and *Office31* datasets, respectively. Figure 5 (a) and (b) illustrate the capability of our method on improving the unknown class identification performance. We observe that the value of $acc_u$ is relatively high at the early training stage. This phenomenon is induced by the model that is not well-trained and prefers to output high entropy values for testing samples. Even though the performance of $acc_u$ drops during the training stage, our method still outperforms ADA and MEADA in terms of $acc_u$ while achieves similar mean accuracy $acc$.

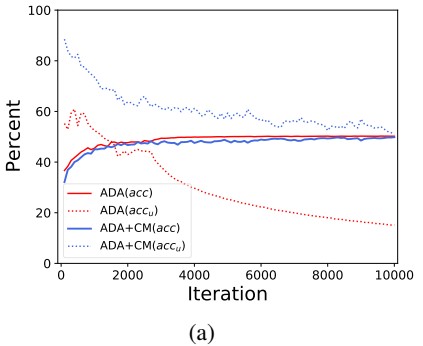
(a)

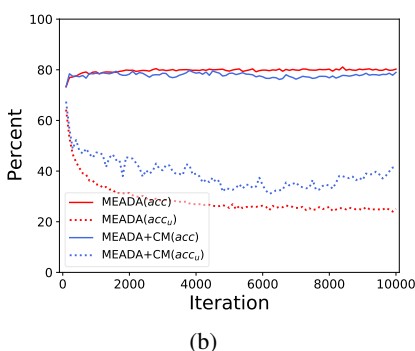
(b)

Figure 5: (a) $acc$ and $acc_u$ convergence of ADA and ours on *Digits*. (b) $acc$ and $acc_u$ convergence of MEADA and ours on *Office31*.

Table 6: Results (%) on *Digits* (ConvNet) under different values of $\alpha$.

| $\alpha$ | | 0.4 | 0.6 | 0.8 | 1.0 | 1.2 | 1.4 | 1.6 |
|---|---|---|---|---|---|---|---|---|
| | $acc$ | 49.65 | 50.6 | 49.56 | 49.71 | 49.18 | 50.18 | 48.64 |
| ADA+CM | $acc_u$ | 49.69 | 51.56 | 47.53 | 52.07 | 52.36 | 48.14 | 56.62 |
| | $hs$ | 38.44 | 41.04 | 38.54 | 39.93 | 39.6 | 39.13 | 40.81 |
| | $acc$ | 50.65 | 52.3 | 51.6 | 51.27 | 51.57 | 50.47 | 51.01 |
| MEADA+CM | $acc_u$ | 46.66 | 37.78 | 42.53 | 46.11 | 42.29 | 51.53 | 48.02 |
| | $hs$ | 39.14 | 36.34 | 37.07 | 38.70 | 37.36 | 41.67 | 40.08 |

## A.6 PARAMETER SENSITIVITY

We analyze the sensitivity of $\alpha$ in Eq. 8 on *Digits* datasets by varying it from 0.4 to 1.6. Table 6 shows performance of our method in terms of $acc$, $acc_u$, and $hs$. Compared ADA and MEADA performance on *Digits* datasets (in Table 2), our method significantly outperforms them with respect to $acc_u$ and $hs$ while obtains comparable results on $acc$, which proves our method is robust to different choices of $\alpha$. We also evaluate the sensitivity of $\mu$ in Figure 3 on *Digits*. As shown in Table 7, With small value of $\mu$, the inference framework would mark many known classes samples as unknown class. With large value of $\mu$, the inference framework will reject many unknown class samples to known classes.

## A.7 PERFORMANCE OF UNKNOWN CLASS IDENTIFICATION AND KNOWN CLASSES CLASSIFICATION

We report the performance of our method and baselines in terms of unknown class accuracy $acc_u$ and known classes accuracy $acc_k$ on *Office-Home* and *PACS* datasets. More details can be found in Table 8 and 9.

## A.8 EXPERIMENTAL RESULTS WITH STANDARD DEVIATION (STD)

We provide STD value for our method and baselines, i.e., ERM, ADA, and MEADA on four datasets, i.e., *Digits*, *Office31*, *Office-Home*, and *PACS* in Table 10, 11, and 12.

Table 7: Results (%) on *Digits* (ConvNet) under different values of $\mu$.

| Metric | $\mu$=0.2 | | $\mu$=0.6 | | $\mu$=0.804 | | $\mu$=1.0 | | $\mu$=1.2 | |
|---|---|---|---|---|---|---|---|---|---|---|
| | *ERM* | *ERM+CM* | *ERM* | *ERM+CM* | *ERM* | *ERM+CM* | *ERM* | *ERM+CM* | *ERM* | *ERM+CM* |
| $acc_k$ | 48.54 | 42.34 | 54.07 | 48.36 | 56.40 | 48.67 | 58.13 | 53.68 | 58.98 | 56.35 |
| $acc_u$ | 47.25 | 69.61 | 25.21 | 53.91 | 13.04 | 53.52 | 5.91 | 38.17 | 1.66 | 29.36 |
| $acc$ | 48.37 | 46.88 | 49.26 | 49.28 | 49.17 | 49.07 | 49.42 | 51.09 | 48.02 | 51.85 |
| $hs$ | 40.82 | 42.68 | 29.69 | 40.28 | 17.97 | 40.15 | 9.77 | 33.88 | 3.09 | 28.98 |

Table 8: Known classes accuracy (%) and unknown class accuracy (%) on *Office-Home* (ResNet18).

| Method | Artistic | | Clip Art | | Product | | Real-World | | Average | |
|---|---|---|---|---|---|---|---|---|---|---|
| | $acc_k$ | $acc_u$ | $acc_k$ | $acc_u$ | $acc_k$ | $acc_u$ | $acc_k$ | $acc_u$ | $acc_k$ | $acc_u$ |
| OSDAP | 44.13 | 67.84 | 51.69 | 69.26 | 40.00 | 63.47 | 52.48 | 66.92 | 47.07 | 66.87 |
| OpenMax | 17.38 | 98.08 | 17.72 | 97.04 | 9.53 | 98.59 | 20.78 | 97.01 | 16.35 | 97.68 |
| ERM | 68.54 | 20.53 | 66.75 | 24.65 | 62.81 | 26.26 | 69.48 | 23.18 | 66.90 | 23.66 |
| ERM+CM | 66.48 | 48.57 | 64.80 | 41.95 | 59.17 | 40.94 | 69.36 | 43.69 | 64.95 | **43.79** |
| ADA | 71.36 | 22.05 | 67.37 | 31.19 | 62.91 | 24.55 | 69.92 | 23.88 | 67.89 | 25.42 |
| ADA+CM | 67.53 | 39.59 | 64.10 | 40.67 | 59.92 | 40.72 | 68.53 | 40.79 | 65.02 | **40.44** |
| MEADA | 71.37 | 22.36 | 66.45 | 31.27 | 62.75 | 25.60 | 69.92 | 23.71 | 67.62 | 25.74 |
| MEADA+CM | 66.63 | 45.28 | 64.43 | 37.84 | 59.74 | 37.71 | 68.82 | 41.28 | 64.90 | **40.53** |

Table 9: Known classes accuracy (%) and unknown class accuracy (%) on *PACS* (ResNet18).

| Method | Art Paint | | Cartoon | | Sketch | | Photo | | Average | |
|---|---|---|---|---|---|---|---|---|---|---|
| | $acc_k$ | $acc_u$ | $acc_k$ | $acc_u$ | $acc_k$ | $acc_u$ | $acc_k$ | $acc_u$ | $acc_k$ | $acc_u$ |
| OSDAP | 54.17 | 49.84 | 41.36 | 51.68 | 38.84 | 54.92 | 28.09 | 41.62 | 40.62 | 49.51 |
| OpenMax | 42.87 | 91.48 | 15.27 | 97.44 | 13.16 | 96.61 | 11.96 | 90.22 | 20.82 | 93.94 |
| ERM | 68.80 | 24.57 | 59.46 | 33.08 | 43.34 | 20.27 | 37.54 | 30.03 | 52.29 | 26.99 |
| ERM+CM | 68.66 | 44.56 | 62.25 | 43.18 | 41.01 | 33.16 | 39.91 | 54.21 | **52.96** | **44.53** |
| ADA | 70.95 | 28.80 | 62.08 | 33.83 | 43.18 | 22.41 | 40.65 | 38.77 | 54.22 | 30.93 |
| ADA+CM | 72.93 | 40.12 | 64.39 | 49.06 | 44.98 | 40.85 | 43.27 | 52.53 | **56.40** | **45.64** |
| MEADA | 70.90 | 28.65 | 62.09 | 33.55 | 43.42 | 22.90 | 39.78 | 40.31 | 54.05 | 31.35 |
| MEADA+CM | 70.45 | 33.36 | 63.76 | 53.74 | 40.25 | 48.79 | 42.89 | 50.57 | **54.34** | **46.61** |

Table 10: Results (%) with STD Value on *Digits* (ConvNet) and *Office31* (ResNet18).

| | | ERM | ERM+CM | ADA | ADA+CM | MEADA | MEADA+CM |
|---|---|---|---|---|---|---|---|
| Digits | $acc$ | 49.17±0.2 | 49.07±0.2 | 50.22±0.1 | 49.71±0.2 | 52.98±0.2 | 51.27±0.1 |
| | $acc_u$ | 13.04±0.1 | **53.52**±0.1 | 15.11±0.2 | **52.07**±0.3 | 29.83±0.2 | **46.11**±0.2 |
| | $acc_k$ | 56.40±0.2 | 48.67±0.2 | 57.24±0.3 | 49.24±0.2 | 57.61±0.3 | 52.30±0.2 |
| | $hs$ | 17.97±0.2 | **40.15**±0.1 | 20.14±0.2 | **39.93**±0.2 | 30.37±0.2 | **38.70**±0.2 |
| Office31 | $acc$ | 79.82±0.3 | 78.30±0.3 | 80.13±0.2 | 78.61±0.2 | 80.26±0.2 | 78.98±0.2 |
| | $acc_u$ | 27.04±0.5 | **37.60**±0.2 | 25.24±0.3 | **34.51**±0.3 | 25.09±0.3 | **41.08** ±0.2 |
| | $acc_k$ | 85.10±0.2 | 82.37±0.2 | 85.62±0.3 | 83.02±0.3 | 85.78±0.3 | 82.77±0.3 |
| | $hs$ | 40.69±0.4 | **51.14**±0.2 | 38.65±0.3 | **48.50**±0.3 | 38.55±0.2 | **54.69**±0.2 |

Table 11: Results (%) with STD value on *Office-Home* (ResNet18).

| Method | Artistic | | Clip Art | | Product | | Real-World | | Average | |
|---|---|---|---|---|---|---|---|---|---|---|
| | *acc* | *hs* | *acc* | *hs* | *acc* | *hs* | *acc* | *hs* | *acc* | *hs* |
| ERM | 65.00±0.3 | 31.07±0.6 | 64.12±0.3 | 35.78±0.6 | 60.53±0.4 | 36.33±0.5 | 66.59±0.3 | 33.92±0.5 | 64.06 | 34.28 |
| ERM+CM | 65.49±0.3 | 52.85±0.4 | 63.37±0.4 | 50.51±0.6 | 58.03±0.3 | 47.25±0.7 | 67.75±0.6 | 52.60±0.4 | 63.66 | **50.80** |
| ADA | 68.29±0.6 | 32.94±0.4 | 65.10±0.4 | 42.09±0.3 | 60.52±0.2 | 34.72±0.3 | 67.04±0.3 | 34.86±0.4 | 65.24 | 36.15 |
| ADA+CM | 66.30±0.4 | 46.68±0.3 | 62.64±0.5 | 49.31±0.4 | 58.72±0.3 | 47.47±0.3 | 66.82±0.4 | 50.47±0.3 | 63.62 | **48.48** |
| MEADA | 68.31±0.3 | 33.29±0.4 | 65.25±0.4 | 42.05±0.5 | 60.43±0.3 | 35.68±0.2 | 67.04±0.3 | 34.65±0.4 | 65.01 | 36.42 |
| MEADA+CM | 65.85±0.4 | 53.22±0.4 | 62.90±0.3 | 48.87±0.2 | 58.36±0.3 | 45.34±0.3 | 67.10±0.2 | 50.77±0.3 | 63.55 | **49.55** |

Table 12: Results (%) on *PACS* (ResNet18).

| Method | Art Paint | | Cartoon | | Sketch | | Photo | | Average | |
|---|---|---|---|---|---|---|---|---|---|---|
| | *acc* | *hs* | *acc* | *hs* | *acc* | *hs* | *acc* | *hs* | *acc* | *hs* |
| ERM | 62.24±0.5 | 38.90±0.4 | 55.34±0.6 | 40.96±0.5 | 39.19±0.4 | 28.89±0.3 | 38.32±0.2 | 35.74±0.4 | 48.77 | 36.12 |
| ERM+CM | 63.52±0.4 | 44.9±0.6 | 57.6±0.4 | 48.31±0.5 | 38.53±0.5 | 30.43±0.6 | 42.52±0.3 | 41.6±0.4 | **50.54** | **41.31** |
| ADA | 62.48±0.6 | 39.02±0.4 | 56.43±0.4 | 41.55±0.5 | 39.03±0.2 | 26.93±0.4 | 40.28±0.6 | 38.13±0.5 | 49.56 | 36.41 |
| ADA+CM | 64.26±0.5 | 42.4±0.6 | 60.41±0.4 | 51.81±0.4 | 42.48±0.3 | 35.18±0.5 | 43.97±0.4 | 42.76±0.7 | **52.78** | **43.04** |
| MEADA | 62.43±0.4 | 38.85±0.3 | 56.1±0.6 | 41.34±0.4 | 38.89±0.7 | 26.43±0.5 | 39.88±0.6 | 38.24±0.4 | 49.33 | 36.22 |
| MEADA+CM | 62.63±0.5 | 41.88±0.6 | 60.03±0.3 | 51.36±0.6 | 41.51±0.4 | 35.76±0.4 | 43.5±0.5 | 41.6±0.5 | 51.92 | **42.65** |

