# OpenReview forum: "CrossMatch: Cross-Classifier Consistency Regularization for Open-Set Single Domain Generalization"
_ICLR.cc/2022/Conference — ICLR 2022 Poster_

### Official Review · Reviewer_46jP · 2021-10-20

**Correctness:** 4
**Technical Novelty And Significance:** 3
**Empirical Novelty And Significance:** 3
**Recommendation:** 5
**Confidence:** 4

**Main Review:**

strengths:
1. this paper proposes to study a new and interesting problem, open-set domain generalization

2. experiments show that CrossMatch improves previous domain generalization methods like ADA (NeurIPS, 2018) and MEADA (NeurIPS, 2020) and obtains impressive results on three object recognition datasets

weaknesses:
1. novelty of the proposed method sounds somewhat incremental compared with prior works in DG (Volpi et al., 2018; Zhao et al., 2020a) and UDA (Liu et al., 2019; Saito & Saenko, 2021)

2. the intuition behind L_{ccr} in Eq.(14) is hard to understand, how does matching the outputs of the one-vs-all classifier and the multi-class classifier benefit the generalization ability? The results of the variant "L_unk^'+L_unk" are missing in Table 4.

3. Also, the results of open-set domain adaptation are missing in the experiments, which are vital and could be considered as the upper bound of this new studied problem setting.


Typos:
1. around Eq.(12), L_{consis} should be L_{ccr}

**Summary Of The Paper:**

This paper proposes a new method called CrossMatch for open-set single domain generalization where only one source domain is available to train the model.
This problem studied here is interesting and sounds reasonable, which is also closely related to open-set domain adaptation and single domain generalization.
In particular, CrossMatch designs a new strategy to generate auxiliary samples for unknown classes and develops a novel consistency regularization to help identify samples from unknown classes in target domains.
Results on several datasets verify that the proposed method achieves impressive results on three widely-used datasets.

**Summary Of The Review:**

This paper proposes to study a new and interesting transfer learning setting called open-set domain generalization and develops a new method (CrossMatch) for this problem. However, CrossMatch is mainly built on several previous methods, e.g., the multi-binary classifier (Liu et al., 2019; Saito & Saenko, 2021), adversarial data augmentation (Volpi et al., 2018; Zhao et al., 2020a), making the overall novelty incremental for ICLR. Thus, I tend to give a "borderline reject" score.

---

> ### Author Response · Authors · 2021-11-23
> **Response to Reviewer 46jP (Part 1)**
>
> We thank the reviewer for providing constructive comments. In the following we provide detailed responses to these questions.
>
> **Q1. Novelty of the proposed method compared with prior works in DG.**
>
> **R1.** In this paper, we proposed a new problem, i.e., open-set single domain generalization (OS-SDG), and designed a new approach to address it. In the following, we would like to clarify the novelty of our proposed method by comparing it with the most relevant works, i.e., open set domain adaptation (OSDA), universarial domain adaptation (UDA), domain generalization (DG), and single domain generalization (SDG).
>
> Existing open set domain adaptation (OSDA) and unversarial domain adaptation (UDA) methods suffer from two major limitations. First, they need to use both source and target domains to train the model. In practice, the target domain is not always available. Second, they only focus on the problem that contains one source domain and one target domain. However, in practice, there might be multiple target domains for evaluation. Our experimental results show that a representative OSDA method, OSDAP, has very limited performance when it’s evaluated in the open-set setting with multiple target domains.
>
> Existing domain generalization (DG) methods also face two major challenges. First, DG methods require multiple source domains for model training. In practice, it would be expensive or even infeasible to collect data from multiple source domains. SDG aims to address this issue by training a model with only a single source domain data and generalizing it to multiple unseen target domains. Second, DG methods require the source and target domains share the same label space, which is usually not guaranteed. Very recently, open domain generalization (ODG) was proposed to address this issue, which assumes source and target domains can have different label space. Clearly, SDG and ODG separately tackle each of the two limitations.
> Different from existing works mentioned above, we propose a more practical yet unsolved problem OS-SDG. It assumes that only one source domain is available for model training, and source and target domains have different label spaces.
>
> **Q2. How does matching the outputs of the one-vs-all classifier and the multi-class classifier benefit the generalization ability? The results of the variant $L_{unk}'$+$L_{unk}$ are missing in Table 4.**
>
> R2. The proposed framework CrossMatch involves two classifiers: (1) multi-class classifier, which aims to classify target samples into corresponding known class; and (2) multi-binary classifier [1], which has proved its ability to distinguish unknown classes from known classes. The inference framework only adopts the multi-class classifier to classify each target sample into a corresponding known class or mark it as unknown class. In order to improve the capability of the multi-class classifier on unknown classes identification, we design a novel cross-classifier consistency regularization, by switching the output of multi-class classifier into one-vs-all format for each class and encouraging it to align the output of multi-binary classifier. As a result, the multi-class classifier can produce similar output distribution with the multi-binary classifier in one-vs-all version for each generated auxiliary sample.
>
> [1] ICCV2021: Ovanet: One-vs-all Network for Universal Domain Adaptation.
>
> For the missing of  $L_{unk}'$+$L_{unk}$. As we mentioned in the paper, $L_{unk}’$ is a baseline, and we aimed to demonstrate that $L_{unk}$ is a better replacement of $L_{unk}'$. Because $L_{unk}'$ is actually not a component of our method, we thought it was unnecessary to implement $L_{unk}'$+$L_{unk}$ for the ablation studies in Table 4.

---

> > ### Author Response · Authors · 2021-11-23
> > **Response to Reviewer 46jP (Part 2)**
> >
> > **Q3. Comparison with open-set domain adaptation.**
> >
> > **R3.** Thank you for the constructive comments. We have compared our method with a representative open-set domain adaptation method (i.e, OSDAP [1]).
> >
> > Results in following two tables show that OSDAP obtains very good performance on unknown class recognition, because it utilizes both the source and target domains for model training. Meanwhile, we find that OSDAP performs worse on common classes identification. One potential reason is that the OSDAP is designed to address open set domain adaptation problem with one source domain and one target domain, so it has limited capability in dealing with multiple target domains. Overall, although our method only uses the source domain for model training, it still performs much better than OSDAP on recognizing samples from known classes.
> >
> > [1] ECCV2018: Open Set Domain Adaptation by Backpropagation.
> >
> > | Dataset  | Metric  |  OSDAP  | OpenMax  | ERM| ERM+CM  |ADA| ADA+CM  |MEADA| MEADA+CM  |
> > |:-------------|:----------:|:-----------:|:------------:|:---------:|:------------:|:-----------:|:------------:|:---------:|:------------:|
> > | Digits |  $acc$  | 41.42  |  42.38  | 49.17  |  49.07 | 50.22   | 49.71 | 52.98  | 51.27  |
> > | Digits |  $acc_u$  | 70.60 |  83.81  | 13.04  |  53.52 | 15.11   | 52.07 | 29.83  | 46.11 |
> > | Digits |  $acc_s$  | 35.59 |  34.40  | 56.40  |  48.67 | 57.24   | 49.24 | 57.61  | 52.30 |
> > | Digits |  $hs$  | 40.46 |  40.67  | 17.97  |  40.15 | 20.14   | 39.93 | 30.37  | 38.70 |
> > |-------------|----------|-----------|------------|---------|------------|-----------|------------|---------|------------|
> >  | Office31 |  $acc$  | 76.51  |  18.19  | 79.82  |  78.30 | 80.13   | 78.61 | 80.26  | 78.98 |
> >  | Office31 |  $acc_u$  | 84.28  |  100.0  | 27.04  |  37.60 | 25.24   | 34.51 | 25.09  | 41.08 |
> >  | Office31 |  $acc_s$  | 75.77  |  10.01  | 85.10  | 82.37 | 85.62 | 83.02 | 85.78  | 82.77 |
> >  | Office31 |  $hs$  | 77.68  |  16.74  | 40.69  |  51.14 | 38.65   | 48.50 | 38.55  | 54.69 |
> >
> > | Method  | Artistic  |  Artistic  | Clip Art  | Clip Art| Product |Product| Real-World  |Real-World| Average  |Average  |
> > |:-------------|:----------:|:-----------:|:------------:|:---------:|:------------:|:-----------:|:------------:|:---------:|:------------:|:------------:|
> > | | $acc$ |$hs$ | $acc$ | $hs$ | $acc$ | $hs$ | $acc$ | $hs$ | $acc$ | $hs$ |
> > | OSDAP | 45.61| 52.35 | 52.78 | 58.82 | 41.45 |47.95 | 53.51| 58.40| 48.34| 54.38|
> > |OpenMax| 22.42| 30.64 | 22.67 | 29.51 | 15.10 | 16.65 | 25.54 | 33.07 | 21.43 | 27.47 |
> > |ERM | 65.00 | 31.07 | 64.12 | 35.78 | 60.53 | 36.33 | 66.59 | 33.92 | 64.06 | 34.28 |
> > |ERM+CM| 65.40 | 52.85| 63.37 | 50.51 | 58.03 | 47.25 | 67.75| 52.60| 63.66 | 50.80 |
> > |ADA | 68.29 | 32.94 | 65.10 | 42.09 | 60.52| 34.72 | 67.04 | 34.86 | 65.24 | 36.15 |
> > |ADA+CM | 66.30 | 46.68 | 62.64 | 49.31 | 58.72 | 47.47 | 66.82 | 50.47 | 63.62 | 48.48|
> > |MEADA | 68.31 | 33.29 | 65.25 | 42.05 | 60.43 | 35.68 | 67.04 | 34.65 | 65.01 | 36.42 |
> > |MEADA+CM| 65.85 | 53.22| 62.90 | 48.87 | 58.36 | 45.34 | 67.10 | 50.77 | 63.55 | 49.55 |

---

### Official Review · Reviewer_3bup · 2021-11-01

**Correctness:** 2
**Technical Novelty And Significance:** 3
**Empirical Novelty And Significance:** 2
**Recommendation:** 5
**Confidence:** 4

**Main Review:**

Strength

- Open-set single domain generalization is a novel, challenging, but practically important problem setting. Figure 1 is a good summarization on relationship between this problem setting and related ones.

- Experimental results with several benchmark datasets show that the proposed scheme makes it possible to extend existing domain generalization methods to detect unknown-class data in the inference phase.


Weakness

- I have several concerns regarding the design of CrossMatch.

  - Do we need to assume that all target domains share the same label space?

  - In Eq. (7), since D(D_u, D_s) > \rho is the only constraint on D_u, it seems that D_u can freely go far away from the source data distribution, if we take supreme with respect to D_u.

  - Although the objective function for a whole training process is defined in Eq. (7), the loss functions in each stage do not follow this definition.

    - Eq. (8) adopts an additional hyperparameter \alpha.

    - Eq. (9) and (10) adopt totally different loss functions, which are L_sdg and L_unk.

  - Considering the motivation of cross-classifier consistency regularization, it would be better to stop gradient for F_b. Is there any reason to minimize L_ccr with respect to F_b?

- Although open-set single domain generalization can be seen as SDG + open-set classification task, the authors only use SDG methods as baselines in the experiments. Naive open-set classification (or out-of-distribution detection) methods should perform good for unknown classes but not for known classes due to domain shift, which highlights the advantage of the proposed method more clearly. I could not judge the significancy of the proposed method from the current experimental results.

- How did the authors tune \mu in the experiments? And is it also used for baseline methods (ERM, ADA, and MEADA)? The accuracy on unknown classes should heavily depend on the setting of \mu.

- Several notations are confusing. I list some in the following.

  - In general, "domain" means a pair of representation and distribution of data [R1], not a dataset.

    [R1] "A Survey on Transfer Learning," IEEE Trans. on KDE, 2009.

  - The arguments of the loss functions sometimes change in the manuscript.

  - In Eq. (7), it should be better to explicitly describe which data distribution is used to take expectation.

  - In Eq. (13), p_b^i (t=1|x) should be equal to p_b^i.

Minor concerns that do not affect my score

- In Table 4, L_unk seems to have more impact on performance than L_ccr, though classifier consistency regularization is only included in the title.



**Summary Of The Paper:**

This study tackles a novel domain generalization task, called open-set single domain generalization, in which a model is to be trained with single source domain but needs to generalize well at unseen target domains that may include unseen classes. To solve this problem, the authors extend single domain generalization methods to be able to learn detection of unseen classes by adopting adversarial data augmentation. Experimental results show that the proposed scheme can facilitates the capability of existing methods on detecting unknown-class data.

**Summary Of The Review:**

Open-set single domain generalization is an interesting and important problem setting. However, the proposed method is not approapriately designed for this problem setting. In addition, the experiments and discussion lack an important related work, which is open-set classification (or out-of-distribution detection). I vote for "weak reject."

---

> ### Author Response · Authors · 2021-11-23
> **Response to Reviewer 3bup (Part 1)**
>
> We thank the reviewer for providing constructive comments. In the following we provide detailed responses to these questions.
>
> **Q1. Assumption on target label space and constraint on D_u.**
>
> **R1.** Actually, there is no need to assume that all target domains share the same label space.  The proposed CrossMatch method could be applied to multiple target domains with different label spaces.
>
> In the experiments, we adopt the per-class average accuracy as the evaluation metric. To facilitate the comparison between our method and baselines across multiple target domains, we employ simple settings in which all target domains share the same label space.
>
> In our proposed problem, i.e., open-set single domain generalization (OS-SDG), we can not access the prior knowledge of unknown classes and guide our model to generate auxiliary samples $D_u$ to mimic the distribution of unknown classes samples. In our framework, we seek to enlarge the distribution of generated auxiliary samples from the original source domain, in order to improve the capability and robustness of the learned model on unknown classes identification. In addition to the constraint $D(D_u, D_s) > \rho$, we have also designed a function $L_{unk}(x)$ to guide the generation of unknown class samples, as clarified in the revised paper.
>
> **Q2. Definitions of loss functions**
>
> **R2.** Thanks for the insightful comments. As the loss functions (7-10) in our original paper may cause confusions, we have updated the definitions and discussions of loss functions in the revised paper.
> We have added the missing hyperparameter $\alpha$ to Eq.(7), and have clarified the maximization stage in our approach by providing more details.
>
>
> **Q3. Considering the motivation of cross-classifier consistency regularization, it would be better to stop gradient for $F_b$**
>
> **R3.** Thank you for your valuable comments. We have evaluated the performance of our method based on MEADA by stopping the gradient for $F_b$. The results are summarized in the following table. We find that the strategy of stopping gradient for $F_b$ obtained worse results on unknown class identification.
>
> | Method | $acc$ |  $acc_s$  |  $acc_u$  | $hs$|
> |:-------------|:----------:|:-----------:|:------------:|:---------:|
> | MEADA |  52.98 | 57.61  |  29.83 |  30.37 |
> | MEADA+CM | 51.27  | 52.30 | 46.11 | 38.70 |
> | MEADA+CM(StopG for $F_b$) | 52.25  | 55.31 | 36.98 | 34.76 |

---

> > ### Author Response · Authors · 2021-11-23
> > **Response to Reviewer 3bup (Part 2)**
> >
> > **Q4. Comparison with open-set classification method.**
> >
> > **R4.** Thank you for your valuable comments. We adopt a representative open-set classification method OpenMax [1] as baseline and evaluate the performance of OpenMax on four datasets, i.e., Digits, Office31, OfficeHome, and PACS. The following tables show the results of OpenMax in terms of overall accuracy ($acc$), accuracy on unknown classes ($acc_u$), accuracy on known classes ($acc_s$), and h-score ($hs$). In summary, OpenMax performs well on unknown class identification, but performs poorly on common classes classification when the domain gap is relatively large among source and target domains.
> >
> > | Dataset  | Metric  |  OSDAP  | OpenMax  | ERM| ERM+CM  |ADA| ADA+CM  |MEADA| MEADA+CM  |
> > |:-------------|:----------:|:-----------:|:------------:|:---------:|:------------:|:-----------:|:------------:|:---------:|:------------:|
> > | Digits |  $acc$  | 41.42  |  42.38  | 49.17  |  49.07 | 50.22   | 49.71 | 52.98  | 51.27  |
> > | Digits |  $acc_u$  | 70.60 |  83.81  | 13.04  |  53.52 | 15.11   | 52.07 | 29.83  | 46.11 |
> > | Digits |  $acc_s$  | 35.59 |  34.40  | 56.40  |  48.67 | 57.24   | 49.24 | 57.61  | 52.30 |
> > | Digits |  $hs$  | 40.46 |  40.67  | 17.97  |  40.15 | 20.14   | 39.93 | 30.37  | 38.70 |
> > |-------------|----------|-----------|------------|---------|------------|-----------|------------|---------|------------|
> >  | Office31 |  $acc$  | 76.51  |  18.19  | 79.82  |  78.30 | 80.13   | 78.61 | 80.26  | 78.98 |
> >  | Office31 |  $acc_u$  | 84.28  |  100.0  | 27.04  |  37.60 | 25.24   | 34.51 | 25.09  | 41.08 |
> >  | Office31 |  $acc_s$  | 75.77  |  10.01  | 85.10  | 82.37 | 85.62 | 83.02 | 85.78  | 82.77 |
> >  | Office31 |  $hs$  | 77.68  |  16.74  | 40.69  |  51.14 | 38.65   | 48.50 | 38.55  | 54.69 |
> >
> > | Method  | Artistic  |  Artistic  | Clip Art  | Clip Art| Product |Product| Real-World  |Real-World| Average  |Average  |
> > |:-------------|:----------:|:-----------:|:------------:|:---------:|:------------:|:-----------:|:------------:|:---------:|:------------:|:------------:|
> > | | $acc_s$ |$acc_u$ | $acc_s$ |$acc_u$ | $acc_s$ |$acc_u$ | $acc_s$ |$acc_u$ | $acc_s$ |$acc_u$ |
> > | OSDAP | 44.13 | 67.84 | 51.69 | 69.26 | 40.00 | 63.47 | 52.48 | 66.92 | 47.07 | 66.87|
> > |OpenMax| 17.38 | 98.08 | 17.72 | 97.04 | 9.53 | 98.59 | 20.78 | 97.01 | 16.35 | 97.68 |
> > |ERM |68.54 | 20.53 | 66.75 | 24.65 | 62.82 | 26.26 | 69.48 | 23.18 | 66.90 | 23.66 |
> > |ERM+CM| 66.48 | 48.57 | 64.80 | 41.95 | 59.17 | 40.94 | 69.36 | 43.69 | 64.95 | 43.79|
> > |ADA | 71.36 | 22.05 | 67.37 | 31.19 | 62.91 | 24.55 | 69.92 | 23.88 | 67.89 | 25.42 |
> > |ADA+CM | 67.53 | 39.59 | 64.10 | 40.67 | 59.92 | 40.72 | 68.53 | 40.79 | 65.02 | 40.44 |
> > |MEADA | 71.37 | 22.36 | 66.45 | 31.27 | 62.75 | 25.60 | 69.92 | 23.71 | 67.62 | 25.74|
> > |MEADA+CM| 66.63 | 45.28 | 64.43 | 37.84 | 59.74 | 37.71 | 68.82 | 41.28 | 64.90 | 40.53|
> >
> > | Method  | Art Paint  |  Art Paint  | Cartoon | Cartoon | Sketch |Sketch| Photo |Photo| Average  |Average  |
> > |:-------------|:----------:|:-----------:|:------------:|:---------:|:------------:|:-----------:|:------------:|:---------:|:------------:|:------------:|
> > | | $acc_s$ |$acc_u$ | $acc_s$ |$acc_u$ | $acc_s$ |$acc_u$ | $acc_s$ |$acc_u$ | $acc_s$ |$acc_u$ |
> > | OSDAP | 54.17 | 49.84 | 41.36 | 51.68 | 38.84 | 54.92 | 28.09 | 41.62 | 40.62 | 49.51 |
> > |OpenMax| 42.87 | 91.48 | 15.27 | 97.44 | 13.16 | 96.61 | 11.96 | 90.22 | 20.82 | 93.94 |
> > |ERM | 68.80 | 24.57 | 59.46 | 33.08 | 43.34 | 20.27 | 37.54 | 30.03 | 52.29 | 26.99 |
> > |ERM+CM| 68.66 | 44.56 | 62.25 | 43.18 | 41.01 | 33.16 | 39.91 | 54.21 | 52. 96 | 44. 53|
> > |ADA | 70.95 | 28.80 | 62.08 | 33.83 | 43.18 | 22.41 | 40.65 | 38.77 | 54.22 | 30.03 |
> > |ADA+CM | 72.93 | 40.12 | 64.39 | 49.06 | 44.98 | 40.85 | 43.27 | 52. 53 | 56.40 | 45.64 |
> > |MEADA | 70.90 | 28.65 | 62.09 | 33.55 | 43.42 | 22.90 | 39.78 | 40.31 | 54.05 | 31.35 |
> > |MEADA+CM| 70.45 | 33.36 | 63.76 | 53.74 | 40.25 | 48.79 | 42.89 | 50.57 | 54.34 | 46.61 |
> >
> >
> > [1] CVPR2016: Towards Open Set Deep Networks

---

> > > ### Author Response · Authors · 2021-11-23
> > > **Response to Reviewer 3bup (Part 3)**
> > >
> > > **Q7. How the threshold $\mu$ affects the performance of the proposed method.**
> > >
> > > **R7.** For the threshold $\mu$, we follow an existing open-set approach [1] and set it to $log(|C_s|)/2$, where $|C_s|$ represents source label space and $log(|C_s|)$ is the maximum value of entropy. We evaluate the performance of our method under different threshold values (varying from 0.2 to 1.2) on Digits dataset. The maximum entropy value is $1.609 = log(5)$. With small values of threshold, the model will tend to assign more samples to the unknown class. Oppositely, the model will classify more samples into common classes. We have included the parameter analysis to the revised paper.
> > >
> > > | Metric  | $\mu=0.2$  | $\mu=0.2$  | $\mu=0.6$  | $\mu=0.6$ | $\mu=0.804$  |$\mu=0.804$  | $\mu=1.0$  |$\mu=1.0$  | $\mu=1.2$  |$\mu=1.2$  |
> > > |:-------------|:----------:|:-----------:|:------------:|:---------:|:------------:|:-----------:|:------------:|:---------:|:------------:|:------------:|
> > > | | ERM| ERM+CM|  ERM| ERM+CM|  ERM| ERM+CM|  ERM| ERM+CM|  ERM| ERM+CM|
> > > | $acc_s$ | 48.54 | 42.34 | 54.07 | 48.36 | 56.40 | 48.67 | 58.13 | 53.68 | 58.98 | 56.35 |
> > > | $acc_u$ | 47.25 | 69.61 | 25.21 | 53.91 | 13.04 | 53.52 | 5.91 | 38.17 | 1.66 | 29.36 |
> > > | $acc$ | 48.37 | 46.88 | 49.26 | 49.28 | 49.17 | 49.07 | 49.42 | 51.09 | 48.02 | 51.85 |
> > > |$hs$ | 40.82 | 42.68 | 29.69 | 40.28 | 17.97 | 40.15 | 9.77 | 33.88 | 3.09 | 28.98 |
> > >
> > > [1] NeurIPS2020: Universarial Domain Adaptation vie Self-Supervision
> > >
> > > **Q6. Several notations are confusing.**
> > >
> > > **R6.** Thank you for the suggestions and for pointing out the typos. We have revised our paper accordingly.
> > >
> > > **Q7. Minor concern: Impacts of L_ccr and L_unk on model performance.**
> > >
> > > **R7.** Both $L_{ccr}$ and $L_{unk}$ are major components in the proposed CrossMatch approach. Table 4 in our paper shows the impact of proposed $L_{unk}$ and $L_{ccr}$ based on MEADA. We have also evaluated these two terms based on ERM and summarized the results in the following table. In this case, results show that $L_{ccr}$ also plays an important role in our approach.
> > >
> > > | Method | $acc$ |  $acc_s$  |  $acc_u$  | $hs$|
> > > |:-------------|:----------:|:-----------:|:------------:|:---------:|
> > > | ERM |  49.17  | 56.40  |  13.04  |  17.97  |
> > > | ERM+$L_{unk}$ | 49.94  | 56.27 | 18.33 | 23.23 |
> > > | ERM+$L_{unk}$+$L_{ccr}$ (ERM+CM) | 49.07  | 48.67 | 53.52 | 40.15 |

---

### Official Review · Reviewer_N4tv · 2021-11-01

**Correctness:** 3
**Technical Novelty And Significance:** 3
**Empirical Novelty And Significance:** 3
**Recommendation:** 8
**Confidence:** 4

**Main Review:**

Pros:

+ If there are unknown classes in the target domain, existing DG methods cannot handle this situation. As a result, existing methods will cause significantly prediction errors on such unknown-class data points in the target domain, making the prediction of a network unreliable. In this paper, they propose a new problem setting and a new method to handle this challenging problem.

+ Introducing multi-binary classifier to the OS-SDG problem seems very interesting, since it may identify the unknown-class region well.

+ This paper is easy to follow. Experiments can partially support the claims made in this paper. A plus should be that the authors also design some baselines to their problem setting, which provides solid baselines to see if the gains obtained by CM are significant.

Cons:

- The presentation should be improved. There are many modules introduced in this paper, however, they are not well-motivated. It is better to explain their intuitions why they can help improve the performance.

- I am not sure if it is necessary to list the contributions in the introduction. Such contributions have been described clearly in intro and abs. It seems that you do not need to restate them.

- Key related works are missing. In the literature, open-set domain adaptation and open-set learning are very relevant topics to your proposed problem setting. They should be carefully reviewed and discussed with your method/setting. In some open-set learning papers, they also consider to generate unknown-class data, which is also a key module of your method.

- In the theory of DG, researchers need to assume the relations between source domains and target domain to ensure that DG can be solved. However, this paper does try any theoretical analysis to their problem setting, which raises my concerns regarding the performance of CM on other datasets. It would be great if some theoretical analysis can be concluded or analysed. I would not like to make this problem can only be addressed by some heuristic methods.

- How many times do you repeat your experiments? I did not see error bar/STD values of your methods. This should be provided to verify that the experimental results are stable.

- One key experiment is missing. Openness experiments should be done to show the performance of your method when unknown classes change.

- Although hs is a new criterion for open-set DA, it is better to include the known acc as well. Besides, I did not see acc_u in Table 3, which should be provided.

- I did not see how the threshold affects the performance of your method. The threshold \mu is a very important hyperparameter. How do you choose \mu in the DG problem?



**Summary Of The Paper:**

In domain generalization (DG), label set of target domain is that of source domains. However, we might meet the unknown classes in the target domain, which will cause significantly prediction error on such unknown-class data points in the target domain. To avoid this issue, this paper formulates a new problem setting: open-set DG, where the label set of target domain contains the label set of source domains. This paper actually considers a more challenging problem: open-set single DG (OS-SDG) that extends the problem setting of DG to a more general situation.

To address this very challenging problem, this paper designs a CrossMatch approach to improve the performance of SDG methods on identifying unknown classes by leveraging a multi-binary classifier. CrossMatch generates auxiliary samples out of source label space by using an adversarial data augmentation strategy. This paper also adopts a consistency regularization on generated auxiliary samples between multibinary classifiers and the model trained by SDG methods, to improve the model’s capability on unknown class identification. Experimental results on benchmark datasets prove the effectiveness of CrossMatch on enhancing the performance of SDG methods in the OS-SDG setting.

In general, this paper contributes a novel problem setting and a validate solution to this setting. Although some motivations are unclear, the contributions of this paper are enough.


**Summary Of The Review:**

In general, considering the significance of the researched problem, this paper might be accepted by the ICLR2022. However, some points should be clarified and strengthened in the revision.


-----POST REBUTTAL-----

The authors have addressed my concerns. Thus, I increase my score from 6 to 8.

---

> ### Author Response · Authors · 2021-11-23
> **Response to Reviewer N4tv (Part 1)**
>
> We thank the reviewer for providing constructive comments. In the following we provide detailed responses to these questions.
>
> **Q1. Improve paper presentation, explain the intuitions of the proposed modules, and revise the contribution part in the introduction.**
>
> **R1.** We thank the reviewer for the valuable comments. We have reorganized the content about the modules in this paper to better illustrate the goal of proposing these modules. We have also removed the contributions in the introduction.
>
> **Q2. Related works on open-set domain adaptation and open-set learning.**
>
> **R2.** Thank you for the constructive comments. We have added these related works in our revised paper. We have also evaluated the performance of representative methods in open-set domain adaptation (i.e, OSDAP [1]) and open-set learning (i.e., OpenMax [2]) in the proposed open-set single domain adaptation setting, as shown in the following two tables.
>
>  We can find that OpenMax fails to correctly classify common classes, but it achieves good performance on unknown class identification, by classifying most of the test samples as unknown class. This phenomenon is caused by the domain shift problem, i.e., OpenMax assumes that source and target domains have similar distributions. The open set domain adaptation method, OSDAP, obtains very good performance on unknown class recognition, because it utilizes both the source and target domains for model training. Meanwhile, we find that OSDAP performs worse on common classes identification. More results can be found in our revised paper.
>
> | Dataset  | Metric  |  OSDAP  | OpenMax  | ERM| ERM+CM  |ADA| ADA+CM  |MEADA| MEADA+CM  |
> |:-------------|:----------:|:-----------:|:------------:|:---------:|:------------:|:-----------:|:------------:|:---------:|:------------:|
> | Digits |  $acc$  | 41.42  |  42.38  | 49.17  |  49.07 | 50.22   | 49.71 | 52.98  | 51.27  |
> | Digits |  $acc_u$  | 70.60 |  83.81  | 13.04  |  53.52 | 15.11   | 52.07 | 29.83  | 46.11 |
> | Digits |  $acc_s$  | 35.59 |  34.40  | 56.40  |  48.67 | 57.24   | 49.24 | 57.61  | 52.30 |
> | Digits |  $hs$  | 40.46 |  40.67  | 17.97  |  40.15 | 20.14   | 39.93 | 30.37  | 38.70 |
> |-------------|----------|-----------|------------|---------|------------|-----------|------------|---------|------------|
>  | Office31 |  $acc$  | 76.51  |  18.19  | 79.82  |  78.30 | 80.13   | 78.61 | 80.26  | 78.98 |
>  | Office31 |  $acc_u$  | 84.28  |  100.0  | 27.04  |  37.60 | 25.24   | 34.51 | 25.09  | 41.08 |
>  | Office31 |  $acc_s$  | 75.77  |  10.01  | 85.10  | 82.37 | 85.62 | 83.02 | 85.78  | 82.77 |
>  | Office31 |  $hs$  | 77.68  |  16.74  | 40.69  |  51.14 | 38.65   | 48.50 | 38.55  | 54.69 |
>
> | Method  | Artistic  |  Artistic  | Clip Art  | Clip Art| Product |Product| Real-World  |Real-World| Average  |Average  |
> |:-------------|:----------:|:-----------:|:------------:|:---------:|:------------:|:-----------:|:------------:|:---------:|:------------:|:------------:|
> | | $acc$ |$hs$ | $acc$ | $hs$ | $acc$ | $hs$ | $acc$ | $hs$ | $acc$ | $hs$ |
> | OSDAP | 45.61| 52.35 | 52.78 | 58.82 | 41.45 |47.95 | 53.51| 58.40| 48.34| 54.38|
> |OpenMax| 22.42| 30.64 | 22.67 | 29.51 | 15.10 | 16.65 | 25.54 | 33.07 | 21.43 | 27.47 |
> |ERM | 65.00 | 31.07 | 64.12 | 35.78 | 60.53 | 36.33 | 66.59 | 33.92 | 64.06 | 34.28 |
> |ERM+CM| 65.40 | 52.85| 63.37 | 50.51 | 58.03 | 47.25 | 67.75| 52.60| 63.66 | 50.80 |
> |ADA | 68.29 | 32.94 | 65.10 | 42.09 | 60.52| 34.72 | 67.04 | 34.86 | 65.24 | 36.15 |
> |ADA+CM | 66.30 | 46.68 | 62.64 | 49.31 | 58.72 | 47.47 | 66.82 | 50.47 | 63.62 | 48.48|
> |MEADA | 68.31 | 33.29 | 65.25 | 42.05 | 60.43 | 35.68 | 67.04 | 34.65 | 65.01 | 36.42 |
> |MEADA+CM| 65.85 | 53.22| 62.90 | 48.87 | 58.36 | 45.34 | 67.10 | 50.77 | 63.55 | 49.55 |
>
> [1] ECCV2018: Open Set Domain Adaptation by Backpropagation
>
> [2] CVPR2016: Towards Open Set Deep Networks
>
> **Q3. Assumptions of DG/SDG/OS-SDG and theoretical analysis.**
>
> **R3.** DG and SDG methods assume that the source and target domains share the same label space. In the proposed problem setting OS-SDG, we assume that source and target domains only share some common classes. Thus, there is still a strong relation between source and target domains in our problem setting. Moreover, SDG can be considered as a special case of OS-SDG. Some theoretical results of DG/SDG could be potentially extended to OS-SDG, although theoretical analysis for DG/SDG, e.g., theoretical guarantee of the unbound risk of generalization, is still a big challenge [3]. Our paper focuses on the new problem setting, OS-SDG, and we propose a new approach to address it. Experimental results on multiple real-world datasets demonstrate the difficulty of this new problem and the effectiveness of our approach. We will investigate the theoretical analysis of OS-SDG in our future work.
>
> [3] arXiv2021:  Generalizing to Unseen Domains: A Survey on Domain Generalization

---

> > ### Author Response · Authors · 2021-11-23
> > **Response to Reviewer N4tv (Part 2)**
> >
> > **Q4. Error bar/STD values of the proposed methods.**
> >
> > **R4.** For the Digits dataset, we repeat the experiment ten times. For Office31, OfficeHome, and PACS, we repeat the experiment five times. The STD values validate that the experimental results are stable. We have added the STD values in the revised paper.
> >
> > **Q5. Openness experiments should be done to show the performance of your method when unknown classes change.**
> >
> > **R5.** Thank you for the valuable comment. Actually, we have already performed openness experiments and reported the results in Figure 4 (a) of the original paper. In the experiments,  we varied the size of known classes from 10 to 60 on the Office-Home dataset,, which is equivalent to varying the size of unknown classes from 55 to 15. Results show that our method consistently outperforms the baseline MEADA in terms of  $hs$ and accuracy on unknown classes ($acc_u$), and meanwhile our method obtains similar results than MEADA in terms of the average accuracy ($acc$).
> >
> > **Q6. Include the known class acc in experiments and also report $acc_u$ in Table 3.**
> >
> > **R6.** We have added known class accuracy ($acc_s$) and unknown class accuracy ($acc_u$) in the tables of our revised paper. As shown in the table below, compared with domain generalization baselines ERM, ADA, and MEADA, our method gets similar performance in terms of known class accuracy ($acc_s$), and obtains much better results in terms of unknown class accuracy ($acc_u$).
> > The newly added baselines OpenMax and OSDAP are open-learning and open-set domain adaptation methods. Different from domain generalization methods, OpenMax and OSDAP require both the source and target domains for model training, so they have very good performance in terms of $acc_u$.
> >
> > | Method  | Artistic  |  Artistic  | Clip Art  | Clip Art| Product |Product| Real-World  |Real-World| Average  |Average  |
> > |:-------------|:----------:|:-----------:|:------------:|:---------:|:------------:|:-----------:|:------------:|:---------:|:------------:|:------------:|
> > | | $acc_s$ |$acc_u$ | $acc_s$ |$acc_u$ | $acc_s$ |$acc_u$ | $acc_s$ |$acc_u$ | $acc_s$ |$acc_u$ |
> > | OSDAP | 44.13 | 67.84 | 51.69 | 69.26 | 40.00 | 63.47 | 52.48 | 66.92 | 47.07 | 66.87|
> > |OpenMax| 17.38 | 98.08 | 17.72 | 97.04 | 9.53 | 98.59 | 20.78 | 97.01 | 16.35 | 97.68 |
> > |ERM |68.54 | 20.53 | 66.75 | 24.65 | 62.82 | 26.26 | 69.48 | 23.18 | 66.90 | 23.66 |
> > |ERM+CM| 66.48 | 48.57 | 64.80 | 41.95 | 59.17 | 40.94 | 69.36 | 43.69 | 64.95 | 43.79|
> > |ADA | 71.36 | 22.05 | 67.37 | 31.19 | 62.91 | 24.55 | 69.92 | 23.88 | 67.89 | 25.42 |
> > |ADA+CM | 67.53 | 39.59 | 64.10 | 40.67 | 59.92 | 40.72 | 68.53 | 40.79 | 65.02 | 40.44 |
> > |MEADA | 71.37 | 22.36 | 66.45 | 31.27 | 62.75 | 25.60 | 69.92 | 23.71 | 67.62 | 25.74|
> > |MEADA+CM| 66.63 | 45.28 | 64.43 | 37.84 | 59.74 | 37.71 | 68.82 | 41.28 | 64.90 | 40.53|
> >
> > | Method  | Art Paint  |  Art Paint  | Cartoon | Cartoon | Sketch |Sketch| Photo |Photo| Average  |Average  |
> > |:-------------|:----------:|:-----------:|:------------:|:---------:|:------------:|:-----------:|:------------:|:---------:|:------------:|:------------:|
> > | | $acc_s$ |$acc_u$ | $acc_s$ |$acc_u$ | $acc_s$ |$acc_u$ | $acc_s$ |$acc_u$ | $acc_s$ |$acc_u$ |
> > | OSDAP | 54.17 | 49.84 | 41.36 | 51.68 | 38.84 | 54.92 | 28.09 | 41.62 | 40.62 | 49.51 |
> > |OpenMax| 42.87 | 91.48 | 15.27 | 97.44 | 13.16 | 96.61 | 11.96 | 90.22 | 20.82 | 93.94 |
> > |ERM | 68.80 | 24.57 | 59.46 | 33.08 | 43.34 | 20.27 | 37.54 | 30.03 | 52.29 | 26.99 |
> > |ERM+CM| 68.66 | 44.56 | 62.25 | 43.18 | 41.01 | 33.16 | 39.91 | 54.21 | 52. 96 | 44. 53|
> > |ADA | 70.95 | 28.80 | 62.08 | 33.83 | 43.18 | 22.41 | 40.65 | 38.77 | 54.22 | 30.03 |
> > |ADA+CM | 72.93 | 40.12 | 64.39 | 49.06 | 44.98 | 40.85 | 43.27 | 52. 53 | 56.40 | 45.64 |
> > |MEADA | 70.90 | 28.65 | 62.09 | 33.55 | 43.42 | 22.90 | 39.78 | 40.31 | 54.05 | 31.35 |
> > |MEADA+CM| 70.45 | 33.36 | 63.76 | 53.74 | 40.25 | 48.79 | 42.89 | 50.57 | 54.34 | 46.61 |

---

> > > ### Author Response · Authors · 2021-11-23
> > > **Response to Reviewer N4tv (Part 3)**
> > >
> > > **Q7. How the threshold $\mu$ affects the performance of the proposed method.**
> > >
> > > **R7.** For the threshold $\mu$, we follow an existing open-set approach [1] and set it to $log(|C_s|)/2$, where $|C_s|$ represents source label space and $log(|C_s|)$ is the maximum value of entropy. We evaluate the performance of our method under different threshold values (varying from 0.2 to 1.2) on Digits dataset. The maximum entropy value is $1.609 = log(5)$. With small values of threshold, the model will tend to assign more samples to the unknown class. Oppositely, the model will classify more samples into common classes. We have included the parameter analysis to the revised paper.
> > >
> > > | Metric  | $\mu=0.2$  | $\mu=0.2$  | $\mu=0.6$  | $\mu=0.6$ | $\mu=0.804$  |$\mu=0.804$  | $\mu=1.0$  |$\mu=1.0$  | $\mu=1.2$  |$\mu=1.2$  |
> > > |:-------------|:----------:|:-----------:|:------------:|:---------:|:------------:|:-----------:|:------------:|:---------:|:------------:|:------------:|
> > > | | ERM| ERM+CM|  ERM| ERM+CM|  ERM| ERM+CM|  ERM| ERM+CM|  ERM| ERM+CM|
> > > | $acc_s$ | 48.54 | 42.34 | 54.07 | 48.36 | 56.40 | 48.67 | 58.13 | 53.68 | 58.98 | 56.35 |
> > > | $acc_u$ | 47.25 | 69.61 | 25.21 | 53.91 | 13.04 | 53.52 | 5.91 | 38.17 | 1.66 | 29.36 |
> > > | $acc$ | 48.37 | 46.88 | 49.26 | 49.28 | 49.17 | 49.07 | 49.42 | 51.09 | 48.02 | 51.85 |
> > > |$hs$ | 40.82 | 42.68 | 29.69 | 40.28 | 17.97 | 40.15 | 9.77 | 33.88 | 3.09 | 28.98 |
> > >
> > > [1] NeurIPS2020: Universarial Domain Adaptation vie Self-Supervision

---

> ### Comment · Reviewer_N4tv · 2021-11-28
> **Thanks for the responses from the authors**
>
> After reading the responses from the authors and the revision, my concerns are addressed well. Thus, I would like to support this paper and increase my score from 6 to 8.

---

### Official Review · Reviewer_zAkR · 2021-11-02

**Correctness:** 4
**Technical Novelty And Significance:** 2
**Empirical Novelty And Significance:** 3
**Recommendation:** 6
**Confidence:** 5

**Main Review:**

Strength
+ OS-SDG is an interesting and realistic problem, and the paper clearly described its difference from existing methods.
+ The proposed method makes sense and is technically sound.
+ The experimental results show the effectiveness of the proposed methods.

Weakness
- One concern is that the author maybe could analyze which parts of their approach decrease the known class's accuracy. This could be interesting to know since in some settings like Table 3, adding CM decreases the accuracy for known classes by about 3%.
- Another thing is that some results in the appendix could move forward to replace the converging figures in Figure 4.


**Summary Of The Paper:**

The paper presents a new task: open-set single domain generalization, where only one source domain is available and unknown classes and unseen target domains increase the difficulty of the task. A new method CrossMatch is proposed to solve this new problem. Firstly, auxiliary examples are generated for unknown classes out of the source classes. Then multi-binary classifiers are used to deal with unknown class identification in domain adaptation. Then, the paper proposes a cross-classifier consistency regularization that minimizes the multi-binary classifier's output and one-vs-all multi-class classifier's output. The experiments show the proposed method could largely improve the accuracy for unknown classes in the target domain.

**Summary Of The Review:**

The paper is well-written and is technically sound and the proposed problem makes sense.

---

> ### Author Response · Authors · 2021-11-23
> **Response to Reviewer zAkR**
>
> We thank the reviewer for providing constructive comments. In the following we provide detailed responses to these questions.
>
> **Q1. The author maybe could analyze which parts of their approach decrease the known class's accuracy.**
>
> **R1.** Thank you for the valuable suggestion. We have performed additional experiments to analyze which part of our approach decreases the known class’s accuracy, and have included the new results and discussions in our revised paper.
>
> In our proposed open-set single domain generalization (OS-SDG) problem, we aim to learn a model from a single source domain, generalize it to multiple unseen target domains, and improve its robustness on identifying target samples from unknown classes. Our CM framework includes two components to tackle these challenges: (1) auxiliary sample generation for unknown classes ($L_{unk}$); and (2) cross-classifier consistency regularization ($L_{ccr}$) for generated auxiliary samples. The first one aims to generate auxiliary samples that are potentially out of the source label space to assist in identifying whether a target sample belongs to known classes or not. The second one aims to help the model learn discriminative feature distribution between known and unknown classes and improve the capability of the multi-binary classifier on unknown class identification.
>
> We have evaluated the influence of $L_{unk}$ and $L_{ccr}$ based on ERM on the Digits dataset. As shown in the following table, we find that when added $L_{ccr}$ into the model, it significantly improves the performance of the model in terms of unknown class’s accuracy ($acc_u$) and $hs$ score, but decreases the results on known class’s accuracy ($acc_s$).
>
> | Method | $acc$ |  $acc_s$  |  $acc_u$  | $hs$|
> |:-------------|:----------:|:-----------:|:------------:|:---------:|
> | ERM |  49.17  | 56.40  |  13.04  |  17.97  |
> | ERM+$L_{unk}$ | 49.94  | 56.27 | 18.33 | 23.23 |
> | ERM+$L_{unk}$+$L_{ccr}$ (ERM+CM) | 49.07  | 48.67 | 53.52 | 40.15 |
>
> **Q2. Some results in the appendix could move forward to replace the converging figures in Figure 4.**
>
> **R2.** Thank you for the suggestion. In the revised paper, we have moved table 5 in the appendix to replace the converging figures in Figure 4.

---

### Decision · Program_Chairs · 2022-01-20

**Decision:**

Accept (Poster)

**Comment:**

The paper presents a new problem: open-set single domain generalization, where only one source domain is available and unknown classes and unseen target domains increase the difficulty of the task. To tackle this challenging problem, this paper designs a CrossMatch approach to improve the performance of SDG methods on identifying unknown classes by leveraging a multi-binary classifier. CrossMatch generates auxiliary samples out of source label space by using an adversarial data augmentation strategy. Then, the paper proposes a cross-classifier consistency regularization that minimizes the multi-binary classifier's output and one-vs-all multi-class classifier's output.

The proposed OS-SDG is an interesting and realistic problem. However, since it is way more challenging, the optimal solution to it remains elusive. Some reviewers think the method might be heuristic and lack theoretical guarantees. Nevertheless, the results are promising and the paper makes a first step toward the challenging OS-SDG problem. Another concern is that the CCR loss needs more ablation studies to further analyze its role. Though the authors have added more explanation of this part, I suggest the authors put more ablation studies in the final supplementary document.

Overall, the paper is novel and interesting.  I would recommend acceptance of this paper given its novelty and impressive performance, but I highly suggest the authors add more ablation studies in the final supplementary, as suggested by the reviewers.